# Light-driven activation of mitochondrial proton-motive force improves motor behaviors in a *Drosophila* model of Parkinson's disease

Yuzuru Imai [1,2]*, Tsuyoshi Inoshita[3], Hongrui Meng[4], Kahori Shiba-Fukushima[3], Kiyotaka Y. Hara[5], Naoya Sawamura [6,7] & Nobutaka Hattori[1,2,3,4]*

Mitochondrial degeneration is considered one of the major causes of Parkinson's disease (PD). Improved mitochondrial functions are expected to be a promising therapeutic strategy for PD. In this study, we introduced a light-driven proton transporter, Delta-rhodopsin (dR), to *Drosophila* mitochondria, where the mitochondrial proton-motive force ($\Delta p$) and mitochondrial membrane potential are maintained in a light-dependent manner. The loss of the PD-associated mitochondrial gene *CHCHD2* resulted in reduced ATP production, enhanced mitochondrial peroxide production and lower $Ca^{2+}$-buffering activity in dopaminergic (DA) terminals in flies. These cellular defects were improved by the light-dependent activation of mitochondrion-targeted dR (mito-dR). Moreover, mito-dR reversed the pathology caused by the CHCHD2 deficiency to suppress α-synuclein aggregation, DA neuronal loss, and elevated lipid peroxidation in brain tissue, improving motor behaviors. This study suggests the enhancement of $\Delta p$ by mito-dR as a therapeutic mechanism that ameliorates neurodegeneration by protecting mitochondrial functions.

[1] Department of Research for Parkinson's Disease, Juntendo University Graduate School of Medicine, Tokyo 113-8421, Japan. [2] Department of Neurology, Juntendo University Graduate School of Medicine, Tokyo 113-8421, Japan. [3] Department of Treatment and Research in Multiple Sclerosis and Neuro-intractable Disease, Juntendo University Graduate School of Medicine, Tokyo 113-8421, Japan. [4] Department of Neurodegenerative and Demented Disorders, Juntendo University Graduate School of Medicine, Tokyo 113-8421, Japan. [5] Graduate School of Nutritional and Environmental Sciences, University of Shizuoka, Shizuoka 422-8526, Japan. [6] Research Organization for Nano & Life Innovation, Waseda University, Tokyo 162-8480, Japan. [7] Faculty of Science and Engineering, Waseda University, Tokyo 162-8480, Japan. *email: yzimai@juntendo.ac.jp; nhattori@juntendo.ac.jp

Parkinson's disease (PD) is a neurodegenerative disorder characterized by selective loss of midbrain DA neurons. *CHCHD2* (mutations of which cause an autosomal dominant form of PD) encodes a mitochondrial intermembrane protein[1]. *Drosophila* CHCHD2 (*dCHCHD2*) knockout flies exhibit PD-like phenotypes in an age-dependent manner, which include dysfunction in motor ability, DA neuron loss, increased oxidative stress and mitochondrial cristae degeneration[2]. Loss of dCHCHD2 and introduction of PD-associated human CHCHD2 mutations destabilize cytochrome c, which transports an electron from complex III to complex IV during oxidative phosphorylation (OXPHOS), leading to a reduction in ATP production and the generation of reactive oxygen species (ROS) owing to the electron leak[2].

Mitochondrial electron leak and proton leak affect ROS generation and mitochondrial coupling activity[3]. The electron leak is thought to primarily occur in complex I and complex III where electrons react with $O_2$, forming superoxide, the primary ROS from mitochondria. Basal proton leak accounts for 20% or more of the standard metabolic rate in different organs[4], and increased proton leak, which lowers ROS production by mild uncoupling through uncoupling proteins (UCPs), is thought to occur as the mitochondrial proton-motive force ($\Delta p$) rises[3]. In addition, the maintenance of mitochondrial membrane potential ($\Delta\Psi m$) by $\Delta p$ is important for ATP generation through complex V, the transporter of materials required for mitochondrial functions, including mitochondrial proteins and substrates for metabolic reactions, and $Ca^{2+}$ homeostasis[5,6].

A halophilic bacteria dR drives unidirectional transmembrane ion transport via light-induced isomerization of retinal[7]. The proton pump activity of dR achieves a maximum effect by green light when dR is expressed in the *Escherichia coli* (*E. Coli*) membrane[8]. Mitochondrial expression of the light-driven proton pump dR partially protected mammalian cultured cells from PD-related neurotoxins targeting complex I, which is presumably caused by increased $\Delta p$[9]. However, evaluating the precise effects of dR on neuronal functions in cultured cells under physiological conditions over a long period of time is difficult.

In this study, by adapting dR to mitochondria in *Drosophila*, we present a potential therapeutic approach to preserve mitochondria from degeneration caused by *CHCHD2* loss. Light-dependent activation of mitochondrion-targeted dR (mito-dR) but not a mito-dR inactive mutant successfully transformed mitochondria from an OXPHOS-dependent powerhouse to a photoenergetic powerhouse, which accordingly reinforced the mitochondrial functions of the nerve terminals in terms of ATP production and $Ca^{2+}$-buffering activity, suppressing ROS generation[10,11]. Moreover, the beneficial effects of dR ameliorated the α-synuclein accumulation, DA neuron loss and elevated brain lipid peroxidation caused by *dCHCHD2* loss. Our findings demonstrate that increased $\Delta p$ by light-driven mito-dR reinforces mitochondrial functions, suppressing ROS generation.

## Results

### Generation of flies harboring photoenergetic mitochondria.
Genes responsible for PD have revealed that mitochondrial degeneration is a key factor for PD etiology. Mutations or loss of the PD-associated gene *dCHCHD2* result in reduced OXPHOS activity and increased ROS production in *Drosophila*[2]. Because PD-associated *CHCHD2* mutations have loss-of-function properties, we used *dCHCHD2* knockout flies as a *Drosophila* model of PD[2]. To regenerate mitochondrial activity in the PD model, we designed photoenergetic mitochondria to be expressed in flies. To exclude the possibility that light irradiation itself stimulates mitochondria, we also constructed a

mutant in which the two key residues that interact with retinal, D104 and K225, are replaced by nonfunctional amino acids, N and A, respectively[10,11]. Wild-type (WT) dR showed a red-tinged bacterial pellet when expressed in *E. coli* (Supplementary Fig. 1a). In contrast, the D104N/K225A (NA mutant, hereafter) mutant lost redness similar to a vector control, confirming that the D104N/K225A mutant lacks retinal-binding activity. Light-dependent proton pump activity of dR WT but not NA mutant was also observed in bacteria cells (Fig. 1a). Both WT and NA dR harboring a mitochondrial target signal (mito-dR) successfully localized in mitochondria in *Drosophila* S2R+ cells (Supplementary Fig. 1b). We expressed mito-dR WT and NA in *dCHCHD2*−/− flies along with normal flies using the GAL4-UAS system and confirmed that the expression levels of the two kinds of mito-dR were similar in both lines (Supplementary Fig. 1c). Because dR shows maximum proton activity at ~ 550 nm wavelength with good penetrance through the fly cuticle[8,12], we irradiated flies expressing mito-dR with 550 nm light at 2 Hz for 12 h per day (Fig. 1b). These flies were fed fly food containing 100 μM all-trans-retinal such that dR activity achieves maximum efficiency.

Loss of *dCHCHD2* decreased abdominal motor neuron terminal $\Delta\Psi m$, which was recovered by mito-dR WT but not the nonfunctional mito-dR NA mutant under light irradiation conditions (Fig. 1c). Moreover, an age-dependent reduction in ATP contents in the whole brain of *dCHCHD2*−/− flies was mildly improved to a normal level by light-activated mito-dR (Fig. 1d). To determine whether ATP production is stimulated in DA neurons in which mitochondria are affected in PD, we targeted the expression of mito-dR and ATP biosensor ATeam in DA neurons using the *Ddc-GAL4* driver. ATeam is a genetically encoded Förster resonance energy transfer-based ATP biosensor optimized for low temperatures[13]. We visualized ATP changes in DA neurons in the adult fly brain in a light irradiation-dependent manner (Fig. 1e, f). Although we did not observe significantly increased ATP production by mito-dR WT in DA neuron cell bodies of all fly groups, ATP production was stimulated in the mitochondria of the axonal terminals projecting to the mushroom body in *dCHCHD2*−/− flies (Fig. 1e, f; Supplementary Fig. 1d).

### mito-dR relieves oxidative stress.
Increased $\Delta p$ by mito-dR could cause reverse electron transport, resulting in high levels of superoxide production and subsequent oxidative stress (Supplementary Fig. 2a)[3,14]. If this situation is the case, constitutive reverse electron transport from complex II to complex I could occur, leading to the downregulation of complex I[15]. However, we could not detect significant changes in the activities and protein levels in complexes I and II, suggesting that reverse electron transport does not occur by mito-dR (Supplementary Fig. 2b, c).

Alternatively, proton leakage back to the matrix bypassing complex V through UCPs, which is termed "mild uncoupling", reduces ROS generation[3]. Mild uncoupling is important for the relief of oxidative stress in DA neurons in the substantia nigra pars compacta (SNc) but not in the ventral tegmental area, which is a less-sensitive area in PD[16]. The intensity of a marker for lipid peroxidation, 4-HNE, was higher in *CHCHD2*-deficient flies than in normal flies as previously reported (Fig. 2a, b)[2]. mito-dR activation by light irradiation suppressed the accumulation of 4-HNE signals (Fig. 2a, b). Nonfunctional mito-dR NA with or without light treatment did not change the 4-HNE signals of *dCHCHD2*−/− flies, indicating that light irradiation itself does not suppress brain lipid peroxidation (Fig. 2c, d).

Generation of mitochondrial ROS was visualized by using mitochondrion-targeted roGFP2-Orp1 (mito-roGFP2), a hydrogen peroxide biosensor[17]. We expressed mito-roGFP2 in DA neurons

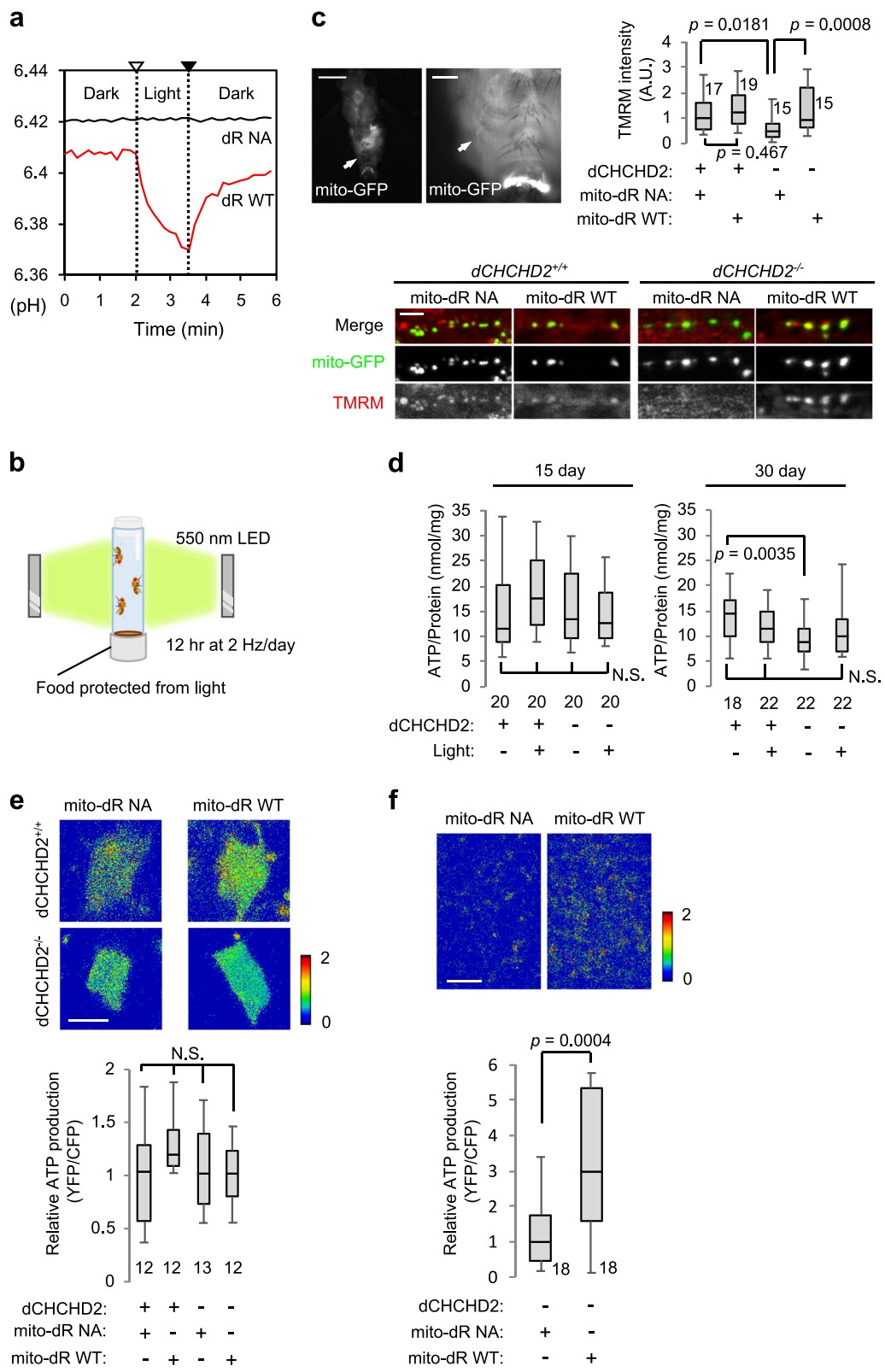

along with mito-dR. As expected, ROS production was increased in the mitochondria of *dCHCHD2*-deficient cell bodies and was suppressed to normal control levels after light activation (Fig. 2e). Light-activated mito-dR also relieved ROS in axonal mitochondria in DA neurons (Fig. 2f). Consistent with the alleviation of oxidative stress, light-activated mito-dR ameliorated the degeneration of mitochondrial cristae caused by *dCHCHD2* loss (Fig. 2g, h)[2].

Given that mito-dR successfully activates mitochondria in the axonal terminals of motor neurons and DA neurons, we next examined the effects of mito-dR in glial cells. Surprisingly, panglial expression of mito-dR WT but not mito-dR NA using the *repo-GAL4* driver in *dCHCHD2*[−/−] flies resulted in early lethality under light irradiation, so that adult flies died by 20 days after eclosion. The expression of mito-dR in the glial

**Fig. 1** Introduction of mitochondrial dR improves ATP production in *dCHCHD2*-deficient flies in a light-dependent manner. **a** H$^+$-pumping activity of dR WT and dR D104N/K225A (NA) mutant in *E. coli*. Open and closed triangles represent light on and off, respectively. **b** Diagram showing light stimulation of mitochondria in flies. Newly eclosed adult files were irradiated with 550 nm LED (7 μW mm$^{-2}$) at 2 Hz for 12 hr every day. Fly food containing 100 μM retinal, which were protected from light with aluminum foil, was changed with fresh food every day. **c** The reduction in ΔΨm in *dCHCHD2$^{-/-}$* neuronal terminals is improved by mito-dR WT but not mito-dR NA. The mitochondria of the abdominal motor neuron terminals of 20-day-old flies visualized with mito-GFP (arrows in upper images, scale bars = 500 μm (left) and 100 μm (right)) were stained with a ΔΨm indicator TMRM (lower images, scale bar = 10 μm). Boxes in the graph indicate the 25th to 75th percentiles, and whiskers represent the maximum and minimum values of the signal intensity of TMRM in the mito-GFP regions. A.U., an arbitrary unit. The numbers of samples analyzed are indicated in the graphs. n = 15–19 flies, Tukey–Kramer test. **d** Whole-head ATP contents in 15- and 30-day-old male flies. Graphs represent ATP contents normalized against the protein levels. The numbers of samples analyzed are indicated in the graphs. n = 18–22 flies, Tukey–Kramer test. N.S., not significant. **e, f** Measurement of relative ATP levels (the ratio of 510–580 nm/440–510 nm emission fluorescence intensity) in the cell bodies **e** and nerve terminals **f** of PAM cluster DA neurons in 30-day-old males expressing mito-dR WT or NA using the fluorescence ATP biosensor ATeam. All flies were treated with light stimulation. Scale bars = 5 **e** and 10 **f** μm. n = 12–13 flies, N.S., not significant by Tukey–Kramer test **e**. n = 18 flies, two-tailed *t* test **f**. Transgenes were driven by *D42-GAL4* **c**, *Da-GAL4* **d** and *R58E02-GAL4* **e, f**. Source data are provided as a Source Data file.

subpopulation using the *nrv2-GAL4* driver, which is active in cortex glia, astrocyte-like glia and ensheathing glia, did not affect the survival of *dCHCHD2$^{-/-}$* flies and stimulated ATP production, whereas simultaneously increasing mitochondrial ROS (Supplementary Fig. 1e)[18]. Glycolytically active glial cells have been shown to produce alanine and lactate to maintain neuronal survival[19]. Our data suggest that excess mitochondrial activation in glia could compromise energy homeostasis and redox status in the glia-neuron metabolic connection.

**mito-dR suppresses α-synuclein accumulation.** SNc DA neurons have an autonomous pacemaking property involving L-type Ca$^{2+}$ channels[16,20]. Mitochondria regulate cellular Ca$^{2+}$ concentrations upon Ca$^{2+}$ influx via the opening of Ca$^{2+}$ channels during synaptic activity whereby the continuous firing of DA neurons is secured. Dysregulation of Ca$^{2+}$ might render SNc DA neurons vulnerable to various stresses, especially oxidative stress, because Ca$^{2+}$ influx to mitochondria stimulates OXPHOS activity, resulting in high ROS generation when antioxidant ability is compromised[16]. As mitochondrial energy production in the axonal terminals of DA neurons was improved in *dCHCHD2$^{-/-}$* flies, we next focused on the Ca$^{2+}$ uptake activity of mitochondria in DA terminals using GCaMP6f and mito-GCaMP6, which are cytosolic and mitochondrial Ca$^{2+}$ indicators, respectively (Supplementary Fig. 3a)[21]. In *Drosophila*, PAM DA neurons, which regulate locomotion and reward, project to the mushroom bodies[22,23]. We electrically stimulated the antennal lobes to monitor the dynamics of cytoplasmic and mitochondrial Ca$^{2+}$ in PAM DA terminals (Supplementary Fig. 3b). Upon mito-dR NA expression, the elevated cytoplasmic Ca$^{2+}$ concentration ([Ca$^{2+}$]$_c$) in DA terminals tended to be higher, and the decay was delayed in the absence of dCHCHD2. In contrast, the increase in the mitochondrial Ca$^{2+}$ concentration ([Ca$^{2+}$]$_m$) was lower, suggesting that Ca$^{2+}$ uptake of mitochondria is impaired in *dCHCHD2$^{-/-}$* DA terminals (Supplementary Fig. 3c). The expression of mito-dR WT in *dCHCHD2$^{-/-}$* flies improved Ca$^{2+}$ uptake to the mitochondria, suppressing the excessive elevation and delayed decay of [Ca$^{2+}$]$_c$ (Fig. 3a). These observations suggest that mito-dR rescues the dysregulation of synaptic activity-dependent Ca$^{2+}$ signaling in *dCHCHD2$^{-/-}$* DA terminals.

In addition to mitochondrial dysfunction, α-synuclein aggregation and accumulation in DA neurons are considered the major causes of PD etiology. Two possible pathological mechanisms between α-synuclein and mitochondria have been proposed. One potential mechanism is mitochondrial damage induced by aggregated α-synuclein[24]. Another is mitochondrial dysfunction leading to α-synuclein aggregation[25,26]. A pathological analysis of patients with *CHCHD2* mutations supports the latter case, and CHCHD2 deficiency or mutations in *Drosophila* also promote

α-synuclein accumulation[27]. The ectopic expression of α-synuclein in DA neurons of *dCHCHD2$^{-/-}$* flies resulted in sarkosyl-insoluble aggregation owing to mitochondrial dysfunction (lane 6 in Fig. 3b and lane 6 in Supplementary Fig. 3d). Importantly, light-activated mito-dR dramatically reduced α-synuclein levels (lane 8 vs. lane 6 in Supplementary Fig. 3d and Supplementary Fig. 3e), which was not owing to DA neuron loss because mito-dR expression with light irradiation was comparable to that without light irradiation (lane 4 vs. lane 2 in Supplementary Fig. 3d). Moreover, light-irradiated mito-dR NA had a null effect on α-synuclein levels, excluding the possibility that light irradiation itself facilitates α-synuclein degradation (lane 6 vs. lane 8 in Fig. 3b).

Consistent with the biochemical changes in α-synuclein, ubiquitinated α-synuclein as well as ubiquitinated puncta were increased in the DA neurons of aged *dCHCHD2$^{-/-}$* flies (3rd vs. 1st row in Fig. 3c). Light-activated mito-dR WT but not mito-dR NA alleviated the accumulation of ubiquitinated protein inclusions, including α-synuclein (4th vs. 3rd row in Fig. 3c).

**Functional improvement in DA neurons by mito-dR.** Age-dependent loss of DA neurons is a pathological feature of PD, and it was also observed in the PPL1, PPM2, and PPM3 clusters of DA neurons in *dCHCHD2$^{-/-}$* flies (*dCHCHD2$^{-/-}$*, mito-dR NA vs. dCHCHD2$^{+/+}$, mito-dR NA in the lower graph in Fig. 4a). The expression of mito-dR WT but not mito-dR NA by the *Ddc-GAL4* driver, which covers most PPM2 clusters and part of PPL1 clusters (Supplementary Fig. 4a), suppressed DA neuron loss (upper graph in Fig. 4a). Although mito-dR was not expressed in most PPM3 DA neurons, a mild rescue was observed in PPM3 neurons, suggesting a non-autonomous effect by neighboring mito-dR-positive neurons (*dCHCHD2$^{-/-}$*, mito-dR WT vs. dCHCHD2$^{-/-}$, mito-dR NA in the lower graph in Fig. 4a). Dopamine production in the heads from flies at 15 days old but not 30 days old was also improved by light-activated mito-dR (Fig. 4b). Consistent with the improvement in DA functions, flight activity, which is regulated by DA neurons[28], was enhanced at 14 days old by light irradiation (left in Fig. 4c). The increased flight event was induced by mito-dR function but not by light irradiation alone because mito-dR NA failed to activate flight behaviors even in the presence of light (right in Fig. 4c). Increased spontaneous locomotion activity of *dCHCHD2$^{-/-}$* flies was also observed with mito-dR WT (Supplementary Fig. 4b). These results suggest that mitochondrial activation of DA neurons by mito-dR enhances motor behaviors.

Mild uncoupling via UCPs reduces ROS generation in mitochondria[3]. Among the UCP family, brain-enriched UCP4 and UCP5 could be involved in PD etiology[16,29–31]. *Drosophila* has three UCP4 homologs in addition to one UCP5 homolog, and only

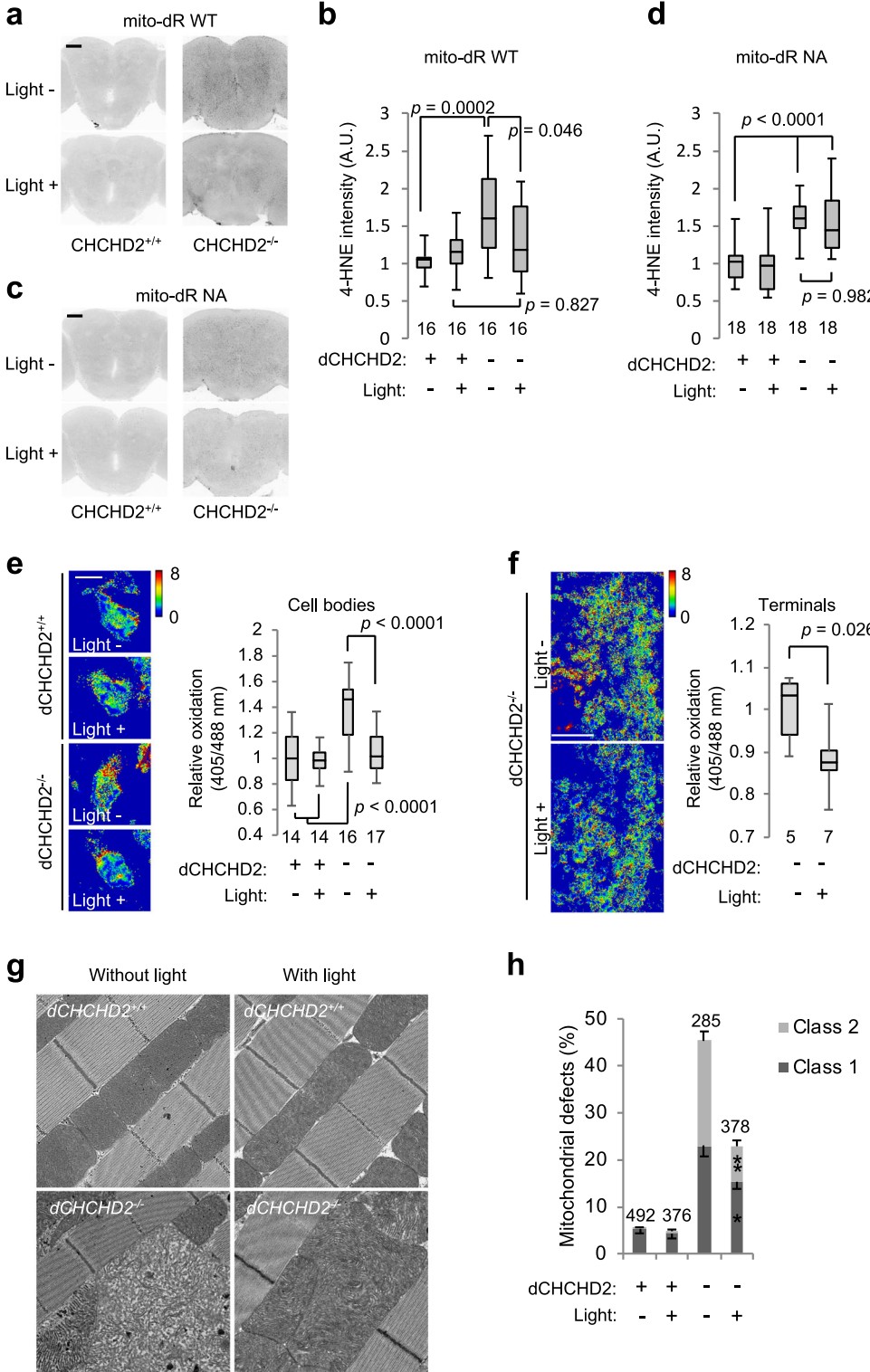

**Fig. 2** Oxidative stress in *dCHCHD2*-deficient DA neurons is alleviated by mitochondrial dR. **a–d** Lipid oxidation by *dCHCHD2* loss is ameliorated by mito-dR WT **a**, **b** but not mito-dR NA **c**, **d**. The whole brain tissues of 15-day-old flies were stained with an anti-4-HNE antibody. Scale bar = 50 μm. The intensities of the anti-4-HNE immunoreactive signals in the whole brain were measured and graphed. $n = 16$–18 flies, Tukey–Kramer test. **e**, **f** Monitoring of mitochondrial redox in the cell bodies **e** and nerve terminals **f** of PAM cluster DA neurons in 14-day-old males using mitochondrial $H_2O_2$ biosensor mito-roGFP2-Orp1. Scale bars = 5 **e** and 10 **f** μm. $n = 14$–17 flies, Tukey–Kramer test **e**. $n = 5$–7 flies, two-tailed Student's *t* test **f**. **g** TEM images of the indirect flight muscle in 14-day-old adult flies expressing mito-dR WT with or without light irradiation are shown. Scale bar = 1 μm. **h** Quantification of mitochondria with abnormal cristae or degenerating mitochondria using a previously reported scoring system as follows[2]: Class 0, normal; Class 1, swirling, fuzzy or dilated cristae; and Class 2, fragmented cristae, and loss of electron density. Mitochondrial defects defined as class 1 and class 2 were counted and are presented as percentages (mean ± s.e.m.). *$p = 0.0012$, **$p < 0.0001$ *vs.* the same classes of *dCHCHD2*$^{-/-}$ without light. $n = 285$–492 from 3 to 4 independent samples. Transgenes were driven by *Da-GAL4* **a–d**, **g**, **h** and *R58E02-GAL4* **e**, **f**. Source data are provided as a Source Data file.

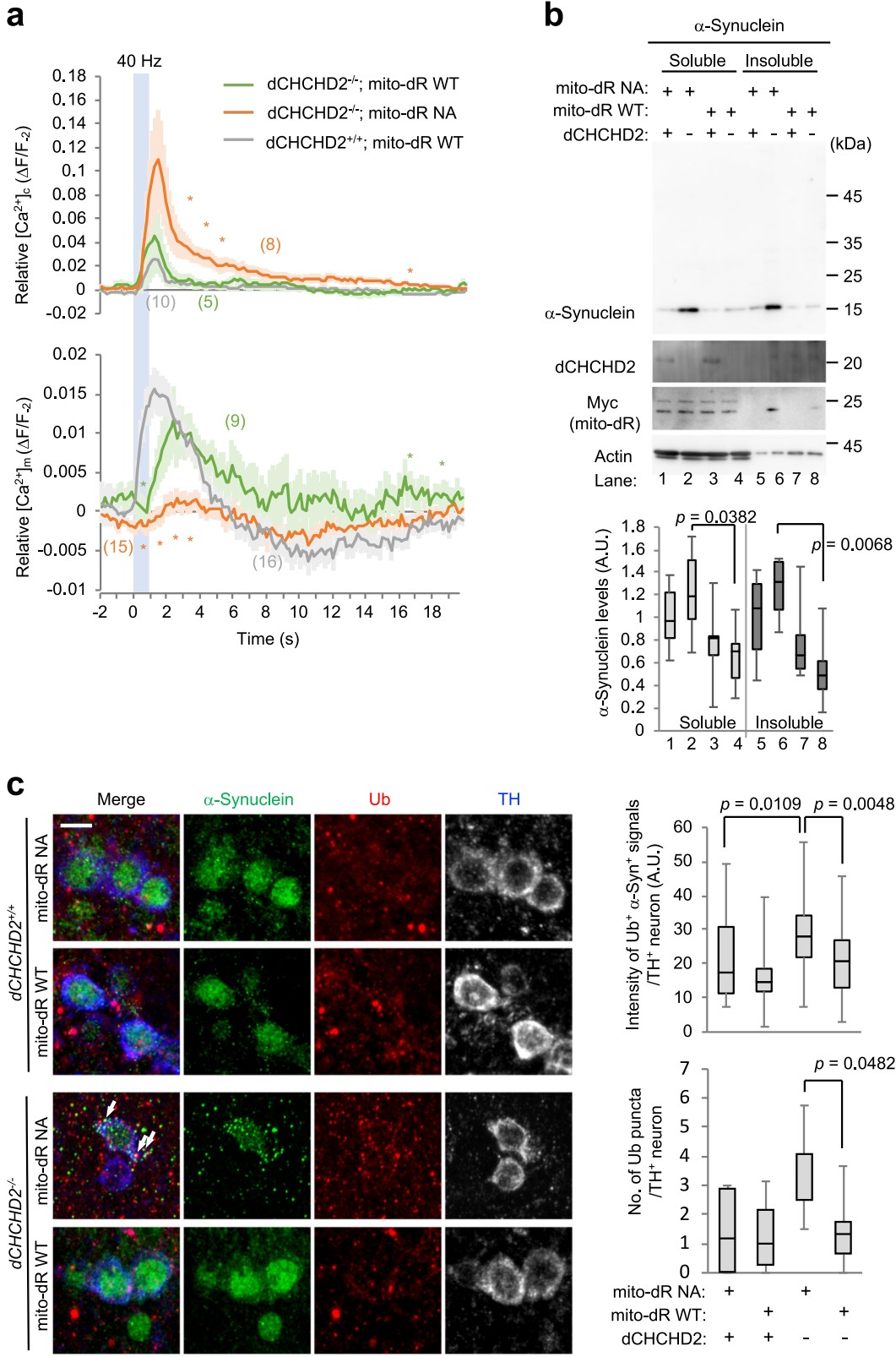

UCP4A has a protective role in PD fly models[31]. Thus, we examined the effects of UCP4A inhibition on the neuroprotective activity of mito-dR. Dopaminergic (DA) coexpression of UCP4A RNAi but not LacZ RNAi with mito-dR weakened the neuroprotective effect on DA neurons (Fig. 4d; Supplementary Fig. 4c) and failed to suppress the decline in motor behavior in aged $dCHCHD2^{-/-}$ flies (Fig. 4e). In contrast, DA coexpression of

UCP4A tended to rescue DA neuron loss in aged $dCHCHD2^{-/-}$ flies expressing nonfunctional mito-dR NA (Supplementary Fig. 4d). However, the coexpression of UCP4A with mito-dR reduced DA neuron survivability in $dCHCHD2^{+/+}$ flies, considering that the normal number of PPM2 DA neurons covered by the $Ddc$-$GAL4$ driver is ~6 (Supplementary Fig. 4d, see also Fig. 4a). These findings suggest that mild uncoupling via UCPs (among which

**Fig. 3** dR improves the mitochondrial functions of *dCHCHD2*-deficient flies. **a** Mitochondrial uptake of $Ca^{2+}$ in PAM DA nerve terminals is improved by mito-dR. Traces (mean ± s.e.m.) of relative fluorescence intensity changes from 2 s before stimulation were graphed. A blue bar indicates 40 Hz electrical stimulations (a set of 10 ms intervals at 5 V and 15 ms duration). *$p < 0.05$, two-tailed Student's *t* test in the integrated value of every s. The numbers of flies analyzed are described in the parentheses of the graphs. GCaMP proteins and mito-dR were driven by *R58E02-GAL4*. **b** mito-dR-mediated mitochondrial activation suppresses α-synuclein accumulation by dCHCHD2 loss. α-synuclein and mito-dR transgenes were coexpressed in DA neurons using the *Ddc-GAL4* driver. Sarkosyl-soluble and insoluble brain extracts from light-irradiated flies expressing mito-dR WT or NA were analyzed by western blotting with the indicated antibodies. Boxplots represent quantitative western blot analysis of α-synuclein. The band intensity of dCHCHD2[+/+], mito-dR NA in each fraction was set to 1. Brain tissues from 30-day-old male flies were analyzed. A.U., arbitrary units. $n = 6$ biological replicates, Tukey–Kramer test. **c** mito-dR-mediated mitochondrial activation suppresses α-synuclein aggregation by dCHCHD2 loss. Arrows indicate α-synuclein (green)- and Ub (red)-positive puncta in PPM1/2 cluster TH-positive neurons (blue). Scale bar = 5 μm. Graphs show the appearance of ubiquitinated α-synuclein signals (upper) and the number of over 5 $nm^2$ ubiquitin-positive puncta in PPM1/2 cluster TH-positive neurons ($n = 32$–44 neurons from 4–5 independent flies, Tukey–Kramer test). Transgenes were driven as in **b**. Source data are provided as a Source Data file.

UCP4A functions dominantly) relieves ROS generation by *dCHCHD2* loss, but chronic expression of UCP4A might have an adverse effect in normal flies.

## Discussion

Mitochondria are the cellular powerhouse in which ATP production is generated by the OXPHOS reaction. A chain of the respiratory complexes I–IV pump up $H^+$ from the matrix, generating the electrochemical potential gradient in the intermembrane between inside and outside. This Δp enables F-type ATPase complex V to synthesize ATP and facilitates mitochondrial transport via translocators such as Tim complexes. Mitochondrial oxidative stress originates from the electron leak from the defective respiratory chain[3] and is considered a major contributor to the pathogenesis of PD.

*dCHCHD2*-deficient flies well recapitulate the PD-like phenotype with age[2]. Because the loss of cytochrome c-binding protein CHCHD2 destabilizes cytochrome c located between complex III and complex IV, leading to electron leakage and excessive oxidative stress, the compensation of respiratory complex I–IV activity seems to be a promising therapeutic target for PD. To overcome the mitochondrial defects caused by OXPHOS dysfunction, we introduced an archaeal proton transporter dR to mitochondria. Light-activated dR increases Δp, which promotes ATP production and probably ΔΨm-dependent transport of materials for mitochondrial functions. In addition, mito-dR suppressed oxidative stress derived from mitochondria, which is likely owing to uncoupling mediated by proton leakage where UCP4A is involved (Fig. 4f)[3,16,31]. Consistent with this observation, the elevation in brain lipid oxidation by CHCHD2 loss was successfully suppressed by mito-dR (Fig. 2a–d). However, brain lipid oxidation in normal flies (*dCHCHD2*[+/+]) was not further reduced. A similar situation was also observed in the cell bodies of DA neurons (Fig. 2e). Although $H_2O_2$ from mitochondria is thought to be the major oxidant species in the brain[32], autooxidation from environmental conditions such as oxygen in the air, short wavelength light, age-dependent impairment in redox regulation and inflammation might cause oxidation of brain lipids and the roGFP2 biosensor as basal oxidant factors. In addition, we observed a subset of glial cells with mildly elevated mitochondrial $H_2O_2$ production when mito-dR was activated in these cells (Supplementary Fig. 1e). These factors may contribute to a null effect of mito-dR in *dCHCHD2*[+/+] flies when mito-dR is ubiquitously expressed.

The vulnerability of SNc DA neurons may arise from their peacemaking activity where $Ca^{2+}$ influx via L-type $Ca^{2+}$ channels are involved. Sustained $Ca^{2+}$ influx incurs a high metabolic cost owing to ATP-dependent $Ca^{2+}$ exclusion as well as mitochondrial oxidative stress owing to the OXPHOS reaction activated by $Ca^{2+}$. Mito-dR seems to resolve this challenge by elevating ATP production and antioxidant activity. Consistent with the

beneficial effects on the mitochondrial functions of DA neurons, dopamine production, locomotor activity and flight behavior, which are regulated by DA neurons, were improved in *dCHCHD2*-deficient flies. In contrast, mito-dR WT did not increase ΔΨm or ATP production in *dCHCHD2*[+/+] flies, suggesting that an overly polarized state does not physiologically occur in mitochondria (Fig. 1c–e). Supporting this idea, the inhibition of F-type ATPase complex V by oligomycin did not prominently increase ΔΨm in human cultured cells[33]. This observation could be explained by uncoupling or proton leak mediated by UCPs, the adenine nucleotide translocase, the glutamate carriers and the permeability transition pore complex[34,35]. *dCHCHD2*[+/+] flies expressing mito-dR WT exhibited a tendency toward a rapid $[Ca^{2+}]_m$ efflux compared with *dCHCHD2*[−/−] flies. This effect might be due to prolonged uncoupling by the $Na^+/H^+$ exchanger (which could potentially activate the $Na^+/Ca^{2+}$ exchanger) or the activation of a putative $Ca^{2+}/H^+$ exchanger such as Letm1 in the energized state[36,37]. These altered $Ca^{2+}$ dynamics in DA neurons might reflect the behavioral changes observed in *dCHCHD2*[+/+] flies.

In conclusion, this study provides 'proof of concept' that Δp maintenance is beneficial for neuroprotection and that the development of proton pumps driven by optogenetic or pharmacogenetic techniques is a potential therapeutic strategy for PD.

## Methods

**Antibodies and plasmids**. The antibodies used in western blot analysis were as follows: anti-dCHCHD2 (1:1000 dilution; in-house[2]), anti-Myc (1:1000; Millipore, clone 4A6), anti-Tim23 (1:2000; BD, clone 32/Tim23), anti-GAPDH (1:2000; Bioss, clone 3E12), anti-actin (1:10,000; Millipore, clone C4), anti-NDUFS3 (1:2000; Abcam, clone 17D95), anti-SDHA (1:1000; GeneTex, GTX101689), anti-UQCRC1 (1:1000; ThermoFisher Scientific, clone 11A51H12), anti-COX IV (1:1000; Abcam, clone 20E8C12), and anti-ATP5A (1:20,000; Abcam, clone 15H4C4). The following antibodies were used for immunohistochemistry: anti-4-HNE (1:500; JaICA, clone HNEJ-2), anti-α-synuclein (1:500; Abcam, clone MJFR1 and 1:500; BD Biosciences, clone 42/α-synuclein), anti-polyubiquitin (1:200; MBL, clone FK2), and anti-dTH (1:500; Abcam, ab76442 and 1:250; in-house[38]). A complementary DNA fragment of myc-mito-dR in the pCMV[9] vector was subcloned into the EcoRI and XbaI sites of the pUAST vector to generate mito-dR WT. The mito-dR NA mutant was generated using QuikChange site-directed mutagenesis kit (Agilent Technologies) with the following primers: dR D104N forward, 5'-GTTGCTGTTGCTCAACC TCTCGCTG; dR D104N reverse, 5'-CAGCGAGAGGTTGAGCAACAGCAAC; dR K225A forward, 5'-GACCTCTCGGCAGCGGTCGGATTCG; dR K225A reverse, 5'-CGAATCCGACCGCTGCCGAGAGGTC. For mito-GaMP6, the mitochondrial targeting sequence of *Drosophila* Hsp60 (CG12101, 1–64 aa) was fused to start codon-removed GCaMP6 by DNA synthesis (GenScript) and cloned into the pUAST vector.

***Drosophila* genetics**. Fly culture and crosses were performed on standard fly food containing yeast, cornmeal, and molasses. Flies were raised at 25 °C unless otherwise stated. Transgenic lines carrying mito-dR or mito-GCaMP6 were generated on the *w[1118]* background (BestGene). All other fly stocks and GAL4 lines used in this study, including *dCHCHD2 null*[2], *UAS-α-Synuclein[LP2]*[39], *UAS-mito-DsRed*[40], *R58E02-GAL4*[22], *UAS-AT1.03NL*[13], *UAS-mito-roGFP2-Orp1*[17], *UAS-UCP4A-3xHA*, and *mb247-DsRed*[23], were obtained from the Bloomington *Drosophila* Stock Center, FlyORF, and Vienna *Drosophila* Resource Center and have been previously

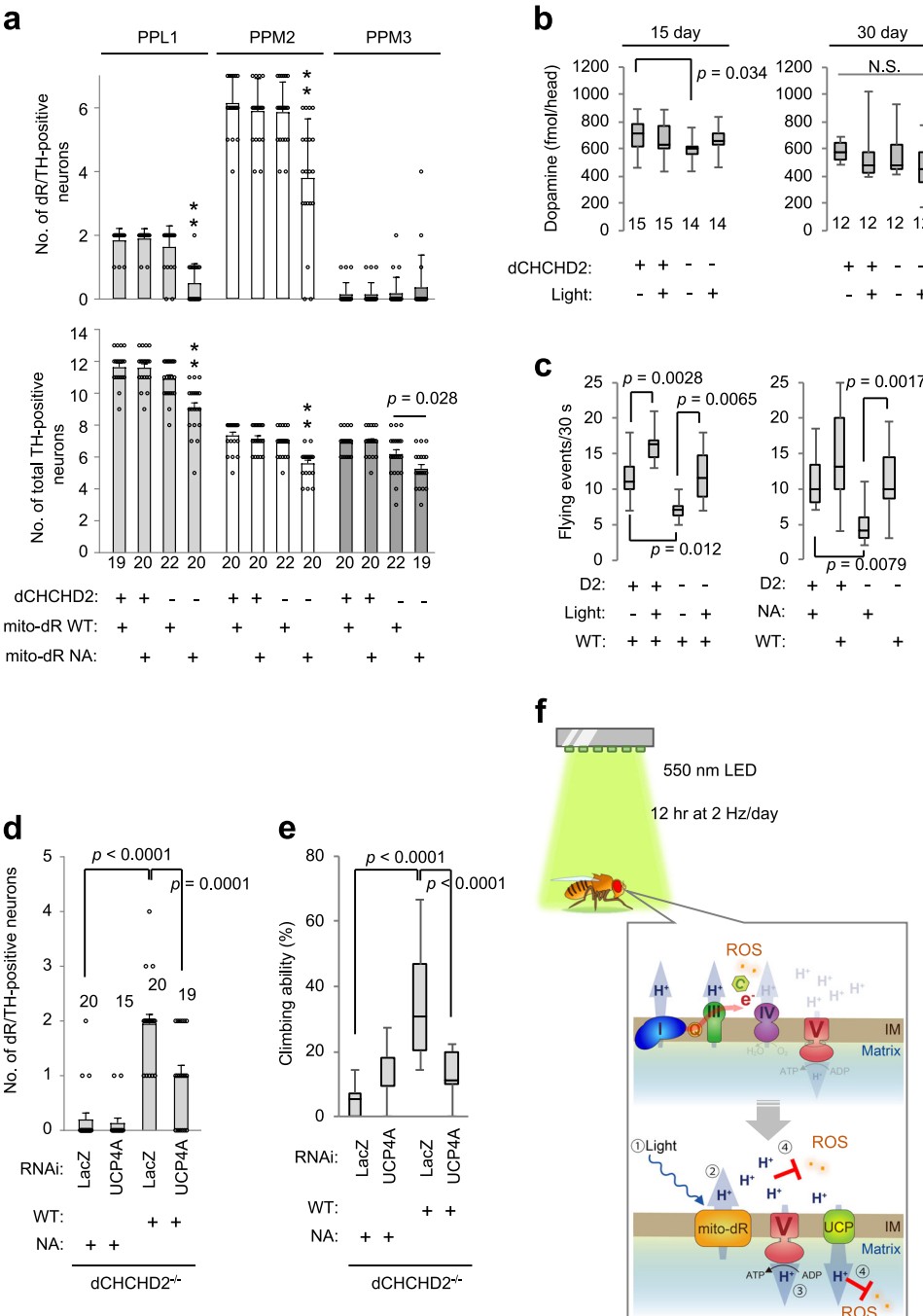

**Fig. 4 Activation of dR rescues DA neurodegeneration in *dCHCHD2*-deficient flies. a** mito-dR WT but not NA rescues the loss of DA neurons in *dCHCHD2*−/− flies. The numbers of dR-positive and total DA neurons in 30-day-old flies were graphed (mean ± s.e.m.). Note that mito-dR was not expressed in most PPM3 DA neurons. **p < 0.0001 vs. the other three groups. n = 19–22 clusters from 10 to 11 flies, Tukey–Kramer test. **b** Dopamine production is partly improved by mito-dR activation. Dopamine levels in the heads were measured in 15- and 30-day-old flies. n = 12–15 flies, Tukey–Kramer test. **c** Reduced flight behavior by dCHCHD2 loss is improved by mito-dR activation. Flying events were recorded for 30 s. D2, dCHCHD2; WT, mito-dR WT; NA, mito-dR NA. n = 10 flies (left) 14 flies (right), Tukey–Kramer test. **d** Inhibition of UCP4A abolishes the neuroprotective role of mito-dR. UCP4A RNAi was coexpressed with mito-dR in DA neurons at 28 °C to enhance RNAi efficiency. The number of PPM2 DA neurons in 28-day-old flies was graphed (mean ± s.e.m.). n = 15–20 clusters from 8–10 flies, Tukey–Kramer test. **e** Inhibition of UCP4A impairs motor activity. The fly groups as in **d** were analyzed at 28 days old. n = 20 trials with 10–12 flies, Tukey–Kramer test. **f** A model of mitochondrial rescue by mito-dR. (Upper) Loss of dCHCHD2 destabilizes cytochrome c, leading to OXPHOS dysfunction, decreased ΔΨm, and ROS production. (Lower) Green light stimulates mito-dR (1), which maintains Δp through pumping out protons from the matrix to the intermembrane space (2), promoting proton influx-dependent ATP synthesis in complex V (3). Increased protons in the intermembrane space quench ROS in the intermembrane space directly and in the matrix by mild uncoupling via UCPs (4). IM, mitochondrial intermembrane. Transgenes were driven by *Da-GAL4* **b** and *Ddc-GAL4* **a**, **c–e**. Source data are provided as a Source Data file.

described. Detailed genotypes used in this study were shown in Supplementary Information.

**Measurement of light-driven proton pumping activity.** pET21d(+)-dR WT and pET21d(+)-dR NA were expressed in the *E. coli* Rosetta (DE3) strain (Novagen). Protein expression was induced by 1 mM isopropyl β-D-thiogalactopyranoside for 3 hr at 37 °C in the presence of 10 μM all-trans-retinal (Sigma-Aldrich). *E. coli* cells were collected by centrifugation at $7000 \times g$ for 2 min and washed three times with 100 mM NaCl. The cells were suspended in 100 mM NaCl at 12 $OD_{600}$. Under dark conditions, 10 ml of the cell suspension was illuminated at 50 mmol $m^{-2} s^{-1}$ using a 300 W halogen projector lamp (JCD100V-300W) through a band path filter within $530 \pm 120$ nm (PBO530-120, Asahi Spectra, Japan). The time course of pH changes in the cell suspension was monitored by pH meter F-72 (Horiba, Japan).

**Light stimulation.** Two 550 nm LED units (250 mm in length and 150 mm in width), which contain 50 bulbs per unit, were regulated by a pulse generator (JW-shop, CON-44RGB-4-V2) and a timer device (MonotaRO, WT-02N-M). The LED units were equipped in an incubator (MEE, CN-40A), and newly eclosed adult flies were irradiated with LED (7 μW $mm^{-2}$) at 2 Hz for 12 hr every day. Fly food containing 100 μM all-trans-retinal, which was protected from light with aluminum foil, was changed with fresh food every day.

**Cell culture and mitochondrial fractionation.** S2R+ cells, which were a kind gift from Dr. N. Yanagawa at Kyoto University, were maintained in Schneider's *Drosophila* medium (Gibco) supplemented with 10% FBS (Gibco) and 1% penicillin–streptomycin (Gibco). S2R+ cells transfected with pUAST-mito-dR along with pAct-GAL4 using HilyMax (Dojindo) for 36 hr were suspended in mitochondrial fractionation buffer (20 mM HEPES, pH 7.3, 220 mM mannitol, 70 mM sucrose, 1 mM EDTA, and protease inhibitor cocktail [Nacalai Tesque]) and were disrupted by passing through a 26-gauge needle twenty times. After centrifugation at $1000 \times g$ for 10 min, supernatants were further centrifuged at $12000 \times g$ for 15 min. Pellets and supernatants were retrieved as mitochondrial and cytosolic fractions, respectively. Both fractions were subjected to western blotting.

**ATP measurement.** Three male fly heads were homogenized in 30 μl homogenization buffer (6 M guanidine, 100 mM Tris-HCl, pH 7.8, and 4 mM EDTA) using a motor-driven pestle. After centrifugation at $20,000 \times g$ for 10 min, the supernatants were diluted at 1:100 and 1:5 with distilled water for ATP and protein quantifications, respectively. ATP contents and protein concentration were measured by CellTiter-Glo luminescent cell viability assay kit (Promega) and protein assay bicinchoninate kit (Nacalai Tesque), respectively.

**Mitochondrial respiratory chain complex activity assay.** The respiratory chain complex activity assays were performed following our previously published protocols[2]. In brief, for the measurement of the complex I activity, NADH dehydrogenase assay buffer (50 mM potassium phosphate buffer pH 7.5, 2.5 mg $ml^{-1}$ bovine serum albumin (BSA), 240 μM KCN, 70 μM decylubiquinone [dUb], 25 μM antimycin A) containing 2 μM rotenone or ethanol was divided into 100 μl aliquots in a microplate and left at RT for 5 min. Complex I activity was monitored by adding 0.4 μg of mitochondrial preparations and 200 μM NADH (2 mM NADH stock) to the reaction mix and following a decrease in absorbance at 340 nm (with reference wavelength at 425 nm). For the measurement of the complex II activity, succinate dehydrogenase assay buffer (25 mM potassium phosphate buffer pH 7.5, 2 mg $ml^{-1}$ BSA, 20 mM potassium succinate, 60 μM 2,6-dichlorophenolindophenol, 25 μM antimycin A, 2 mM KCN, 2 μM rotenone) with or without 10 mM malonic acid was divided into 100 μl aliquots in a microplate and left at RT for 5 min. Complex II activity was monitored by adding 0.4 μg of mitochondrial samples and 100 μM dUb to the reaction mix and following a decrease in absorbance at 600 nm.

**Dopamine measurement.** Three male fly heads were dissected and homogenized in 30 μl 0.007% perchloric acid containing 2 mM EDTA using a motor-driven pestle. Dopamine levels in head extracts were determined by HPLC coupled to electrochemical detection (ESA, Coulochem III) using a mobile phase containing 50 mM citric acid, 50 mM sodium dihydrogen phosphate, pH 2.5, 0.1 mM EDTA, 4.4 mM 1-heptanesulfonic acid, 2.2% (vol/vol) acetonitrile, and 5.3% (vol/vol) methanol.

**Whole-mount immunostaining.** Brain tissues fixed with 4% paraformaldehyde/PBS solution were washed three times with PBS containing 0.3% Triton X-100 (PBS/Tx), blocked with 1% normal goat serum, and stained with the indicated antibodies overnight at 4 °C. After washing with PBS/Tx three times and subsequent incubation with secondary antibodies conjugated to fluorescent dye, tissues were mounted on side glasses using Fluoro-KEEPER antifade reagent (Nacalai Tesque). Images were taken using laser-scanning microscope systems (Leica, TCS-SP5 and Zeiss, LSM880 with Airyscan).

**Transmission electron microscopy (TEM) analysis.** TEM images were obtained using an electron microscope (Hitachi, HT7700) at the Laboratory of Ultrastructural Research of Juntendo University.

**TMRM, ATP, and ROS imaging of neurons.** For tetramethylrhodamine methyl ester (TMRM) staining, the ventral muscles containing the neuromuscular junctions in the abdomen were incubated in 500 nM TMRM (Thermo Fisher Scientific)/HL-3 for 5 min and were washed twice with HL-3. TMRM and mito-GFP signals in the abdominal motor neuron terminals were obtained by a laser-scanning microscope system (Leica, TCS-SP5). TMRM signals in the region of interest set by the binarized mito-GFP images were measured using ImageJ-Fiji software. For imaging of ATP production, male brain tissues dissected in HL-3 containing 5 mM $CaCl_2$ were observed using laser-scanning microscope systems (Leica, TCS-SP5 and Zeiss, LSM880 with Airyscan). To measure the ATP synthesis rate, we observed ATeam images by using a 405 nm excitation laser, and the ratio of 510–580 nm/440–510 nm emission light intensity was calculated[13]. For ROS imaging, a 405 nm or 488 nm excitation laser was used to record roGFP2 signals and calculate the intensity ratio of 405 nm/488 nm. Each imaging session was carried out within 10 min after dissection, and the obtained images were processed using Zen software (Zeiss). The intensities of the ATeam (440–510 nm emission) or roGFP (488 nm excited images) images were thresholded, and values below the threshold were set to 'not a number (NaN)'.

**$Ca^{2+}$ imaging of DA neurons.** Each fly head was held in a hole (1-mm in diameter) in a plastic plate (Fisherbrand, Cat. No. 12–547) with nail polish, and the mouth parts and the cuticles around the antennae were removed by tweezers. $Ca^{2+}$ imaging of brain DA neurons expressing mito-GCaMP6 or GCaMP6f was carried out in $Ca^{2+}$-free HL-3 using an Eclipse FN1 microscope (Nikon) equipped with an electrical stimulation setup containing a SEN-3401 (Nihon koden) and SS-104J (Nihon koden). The lateral side of the antennal lobes was stimulated using a glass electrode (Warner Instruments, Cat. No. W3 64-0792) with a 100-μm tip diameter made by an electrode puller (Sutter Instrument, P-97). GCaMP images were recorded for 1 min (1 frame/0.03 s), and 40 Hz electrical stimulation (5 V with 15-ms duration 10-ms intervals) for 1 s was performed in the antennal regions, as shown in Supplementary Fig. 3b 10 s after the recording. Each imaging was carried out within 10 min after dissection. $Ca^{2+}$ imaging data were processed using NIS-Elements software (Nikon, Ver. AR-4.40.00), ImageJ-Fiji and Excel (Microsoft 2010).

**Sequential extraction of α-synuclein and polyubiquitin.** The biochemical fractionation was performed following our previously published protocols[41]. In brief, 12 male fly heads were homogenized by buffer A (1% sarkosyl, 10 mM Tris-HCl [pH 7.4], 0.8 M NaCl, 1 mM EGTA, 10% sucrose) with protease inhibitor and phosphatase inhibitor cocktails (Nacalai) and sonicated for 3 min on ice. After centrifugation at $1000 \times g$ for 10 min at 4 °C, the supernatants were transferred to ultracentrifuge tubes. After ultracentrifugation at $113,000 \times g$ for 20 min at 4 °C, the supernatants were collected as sarkosyl-soluble fractions. The remaining pellets were washed with 500 μl buffer A, dissolved in 10 μl 3 × SDS buffer and sonicated for 2 min (sarkosyl-insoluble fraction). Both fractions were subjected to western blotting, and the band intensity was analyzed using ImageJ-Fiji software.

**Behavioral assays.** For the flying assay in Fig. 4c, twenty adult male flies at 14 days old were placed in individual vials (93 mm height × 350 $mm^2$ area) at noon and left at rest for 20 min. Flies were dropped down to the bottom by gently tapping, and flight events during 30 s were recorded. Locomotor behavior in Supplementary Fig. 4b was recorded in polycarbonate tubes containing light-shielded fly food with 100 μM retinal using the DAM system (Trikinetics)[42]. The behavior of single male flies preconditioned at 25 °C under a 12-hr light:dark cycle condition for 14 days was individually recorded for 1 additional day. Light irradiation was performed during the light period. For a climbing assay in Fig. 4e, vials (25 mm diameter × 180 mm height) containing 20–25 flies were tapped gently on the table and left standing for 18 s. The number of flies that climbed at least 60 mm was recorded[43].

**Statistics and reproducibility.** Error bars in bar graphs represent the mean ± SEM. Boxes of whisker plots indicate the 25th to 75th percentiles, horizontal lines in the boxes indicate the 50th percentile, and whiskers represent the maximum and minimum values. Statistical analysis was performed by using JMP 11.0.0 (SAS Institute Inc.). Two-tailed Student's *t* test or one-way repeated-measures analysis of variance was used to determine significant differences between the two groups or among multiple groups, respectively, unless otherwise indicated. If a significant result was determined using ANOVA ($p < 0.05$), the mean values of the control and specific test groups were analyzed using a Tukey–Kramer test. Data distribution was assumed to be normal, but this was not formally tested. Abnormal mitochondria (Fig. 2h) and DA neurons (Fig. 4a, d) were counted and graded in a blinded manner by TI and HM. Blinding was not performed in other experiments.

**Reporting summary**. Further information on research design is available in the Nature Research Reporting Summary linked to this article.

## Data availability

All data generated or analyzed during this study available from the authors. The source data underlying all figures are provided in Supplementary Data 1. Full blots are shown in Supplementary Information.

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

## Acknowledgements

We thank Drs. H. Imamura, H. Tanimoto, L.J. Pallanck, T. Riemensperger, M. Saitoe, T. Wang for providing materials; Dr. T. Arano for the initial experiment; and Drs. J. Ogata, C. Cui, Y. Hirono-Hara, and Y. Aoki for their technical assistance. This study was supported by Grants-in-Aid for Scientific Research (16K09679 to T.I., 16K19525 to H.M., 17H04049 to Y.I., 15H04842 to N.H.) from JSPS in Japan and Takeda Science Foundation (to Y.I.); JST-Mirai Program, Japan (JPMJMI17EJ to K.Y.H); and Sumitomo Foundation (to Y.I.), in addition to partial support by a grant from Otsuka Pharmaceutical (N.H. and Y.I.).

## Author contributions

Y.I. conceived the idea, designed and coordinated the research. T.I., H.M., and K.S-F. performed the experiments. Y.I., and T.I. analyzed the data. Y.I. wrote the manuscript. K.Y.H., N.S., and N.H. contributed to logistic support and resources.

## Competing interests

The authors have no conflicts of interest to declare.
