## [Peer Review File · Communications Biology]

Reviewers' comments:

Reviewer #1 (Remarks to the Author):

Imai et al. have used an in vivo drosophila model to investigate the mitochondrial mechanism in a mitochondrial model of PD. Furthermore they use an optogenetic method (overexpression of a light driven proton pump) as a proof of principle to show that modulation of proton transfer/leak during respiration is 1) possible in living flies and 2) is able to alleviate certain PD phenotypes namely electron leak and ROS.

It is novel for several reasons: 1) The gene CHCHD2 is a relatively novel PD gene and little is known about the underlying biology 2) the work is performed in vivo 3) for the first time provide a mitochondrial mechanism that could be exploited as an entry point for therapy. Of course much more work is needed to understand how to translate this to the clinic.

The study is well designed. The strains of drosophila include WT, mutant and then with either functional DR or non-functional DR to prove that the effects are due to the light stimulation. It is not entirely clear whether these expression constructs are somatically expressed, organ specific or neuron specific. In the results some experiments appear to be in flies where the constructs are expressed in the whole body and then later in DA neurons. In the methods the drosophila genetics explains how they generated the flies. The manuscript would be improved by making clear which figures and sub-figures were using which flies.

Please give a more detailed description of how the calcium imaging was performed including an explanation of how the fly brains were dissected for the experiments.

It is not clear whether the statistical analysis refers to statistical n as sole number of flies and/or whether the experiments were repeated on different days or not. We are also concerned about the different numbers of flies per group (n=8-12). This could bias the statistics. Could the authors comment on their statistical approach?

It is interesting to see that the CHCHD2 aged flies have significantly less dopamine but the light activation is not able to alleviate this effect significantly. Does this mean that the mitochondrial effect/damage has already been triggered and initiated a DA specific response? Could the authors comment/discuss why they think the DA phenotype is not altered by the light but the flight and TH neuron count effect is?

It is also interesting that UCP4A seems to be involved in the protection mechanism. It would be very interesting to see whether UCP4A stimulation (or mild uncoupling) in combination with light stimulation could alleviate the DA deficit. Or at least the authors could allude to their hypothesis why mitochondrial targeted treatments may not rescue all DA specific phenotypes.

Minor comments:

Concerning fly age. All flies were tested within a range 30 days or 14/15 days, which seems reasonable. In Figure 4B Imai et al investigate the effect of age at 3 and 15 days old. At least three time points to draw a linear relationship would greatly support the claim of age dependency. The authors refer to Figure 1F when discussing age dependency but here there is only one age group. It is confusing.

Fig 1B: The pH scale bar is a bit confusing and an axis scale bar would be more helpful.

Fig 1C: kDa missing for TIM23 and GAPDH Western blots.

Fig 1D: kDa missing for CHCHD2 and actin.

General. For the box plots the lines defining the axes are missing.

General. Formatting of the figures, especially the graphs could be improved.

Reviewer #2 (Remarks to the Author):

In this manuscript "Light-driven mitochondria relieve a Drosophila model of Parkinson's disease", Imai et al., show that driving the activity of a mitochondria-targeted proton transporter (dR) using light, rescues some of the phenotypes associated with the loss of the PD-associated mitochondrial gene CHCHD2. Specifically, they show that light-induced activation of dR channel can, in some cases elevate ATP levels, as well as decrease mitochondrial peroxide production and restore Ca²⁺-buffering activity of mitochondria—all of which are defective in CHCHD2 null animals. In addition, they show that activation of mito-dR decreases α -synuclein load in DA neurons and restores DA production and motor activity in CHCHD2 flies.

The data are provocative and the manuscript is clearly written, although I had some issues with data presentation which I have outlined below. The authors convincingly show that activation of the dR channel in CHCHD2 null animals is phenotypically protective. However, data regarding the mechanism by which this protection is conferred are less compelling. It is difficult to understand which of the various changes that occur upon activation of dR are responsible for the protective effects, a problem the authors criticize in cultured cells (lines 66-67). In addition, there are conceptual issues as well as missing experimental controls that are needed to convincingly make their point regarding the mechanism of dR activity-dependent rescue in DA neurons.

Major comments:

- 1) A major question throughout, is why dR expression has different effects in the CHCHD2^{+/+} flies versus the CHCHD2^{-/-} flies. If dR activation does indeed act through enhancing Δp in vivo (which is not shown), why do we not see any effects of this—either effects from the increased Δp , or effects due to adaptation to prolonged increases in Δp --in the control animals? This question is evident in multiple cases such as Figure 1f, where activation of dR in dCHCHD2 ^{+/+} animals has no effect on net ATP levels, Figure 3 where dR activation has no effect on Ca²⁺ in dCHCHD2 ^{+/+} animals. This needs to be addressed experimentally, or at the very least, discussed extensively.
- 2) Figure 1f: There is no significant change in ATP production in dCHCHD2 ^{+/+} whole flies as measured by ATP/protein. However, when expressed under a DA driver (Fig 1g), there is an increase in FRET signal in Ateam. These two results are inconsistent and should be reconciled experimentally.
- 3) Figure 2e: Why is ROS not decreased in wt (dCHCD2 ^{+/+}) animals?
- 4) Figure 3a: Changes in [Ca²⁺]_c and [Ca²⁺]_m in dCHCD2 ^{+/+} expressing mito-dR needs to be shown. Is the rescue back to wild-type levels?
- 5) Line 177: α -synuclein aggregation in CHCHD2 ^{-/-} animals. Data should be shown.
- 6) Figure 3b: Are total α -synuclein levels in CHCHD2 ^{-/-} and CHCHD2 ^{+/+} animals expressing mito-dR, equivalent?
- 7) Are changes in ATP production, lipid peroxidation and axonal ROS all occurring in the same neuronal populations in dCHCD2^{-/-} animals? If so, this needs to be better indicated?
- 8) Figures are difficult to read and sloppily put together. For instance, in Figure 3b it is difficult to see what the bar and whisker plots correspond to.

Minor comments:

1) Figure 2d: Does expression of the channel affect 4-HNE levels? 4-HNE levels look lower in dCHCD2 -/- expressing NA vs those expressing mito-dR.

Reviewer #3 (Remarks to the Author):

In the present manuscript entitled "Light-driven mitochondria relieve a Drosophila model of Parkinson's disease", the authors described the characterization of a model in which light-driven proton transporter, Delta-rhodopsin (dR), was targeted to mitochondria to enhance its function. The authors aim to use this reagent to suppress disease phenotypes in Drosophila CHCHD2 loss of function model of Parkinson's disease (PD).

In the Introduction section the authors present an overview on the association of mitochondrial dysfunction in Drosophila CHCHD2 knockout flies, which exhibit PD-like phenotypes. They also highlight the importance of a balance maintenance of mitochondrial membrane potential, proton-motive force, ROS and Ca²⁺ homeostasis in ensuring proper mitochondrial function.

In the Results section, the authors begin with the generation of bacterial constructs of mitochondrial targeted dR, mito-dR WT and mito-dR NA mutant, and demonstrated light-dependent proton pump activity of dR WT but not dR NA in bacterial cells.

Subsequently, the authors generate transgenic flies carrying mito-dR WT as well as mito-dR NA and analyse the phenotypes of their expression in both dCHCHD2 mutant and wild type flies. Light-dependent activation of mito-dR WT but not mito-dR NA inactive mutant enhanced ATP production, Ca²⁺-buffering activity and suppressing ROS generation, specifically in the nerve terminals of adult fly brains.

The authors continue to demonstrate that improved mitochondrial functions achieved via mito-dR expression could suppress α -synuclein accumulation and loss of DA neurons in dCHCHD2 knockout flies.

Finally, the authors demonstrated that inhibition of uncoupling protein, UCP4A, abolished neuroprotective role of mito-dR in dCHCHD2 mutants.

In the Discussion sections, the authors summarize their finding of expression of an archaeal proton transporter to suppress mitochondrial defects caused by OXPHOS dysfunction in dCHCHD2 mutant. They highlighted the "proof of concept" that maintenance of proton-motive force is beneficial for neuroprotection and might have therapeutic potential for PD.

This manuscript is original because it characterizes a novel reagent in which a light-dependent proton transporter was targeted specifically at the mitochondria to enhance its function. In addition, this manuscript also demonstrated the successful use of several optogenetic reagents such as ATP (Ateam), ROS (roGFP2-Orp1) and Ca²⁺ (mito-GCaMP6).

This is an interesting study, with some significant observations and conclusions. However, it is poorly written and overly complicated, paradoxically, even though both the text and figures are complicated, the work is not described in sufficient detail.

As mentioned above the manuscript is poorly written. Specifically, it lacks attention to details, hence making reading difficult. Abbreviations were introduced abruptly in the Introduction, for example:

$\Delta\Psi_m$, Δp and UCPs.

The lack of attention to detail is most obvious in describing experimental designs and data presentation. Specifically, genotypes were not presented in the Figures, Figure Legends nor explained in the text. For example: detection of a-synuclein accumulation in mito-dR expression in dCHCHD2 mutant background (Figure 3b, c), in nowhere was UAS a-synuclein expression described.

Overall, the characterisation of mito-dR transgenic lines is insufficient. It also lacks experimental detail on how mito-dR WT and NA transgenic lines were generated. The reader could only assume they were UAS lines. The author should demonstrate if light-activated mito-dR has an effect on $\Delta\Psi_m$, with a mitochondrial potential indicator, Tetramethylrhodamine, methyl ester (TMRM).

As such, the difficulty in understanding the interpretation of the results obtained has hampered the judgment on the conclusions drawn by the authors. Therefore, additional experiments and inclusion of missing experimental details would be required to support this work.

Specific comments:

- 1) In Figure 1d, the author should specify which tissue or if the whole flies were used in the Western blot.
- 2) In Figure 1f, were the brain ATP contents extracted from whole heads or brain only?
- 3) Figure 1g is poorly described. Was YFP/CFP ratio measured in cell bodies of PAM cluster of DA neurons?
- 4) In Figure 2e, f, and Supplement Figure 3, was it a typo, PAM neurons instead of DAM?
- 5) In Figure 3, genotypes of flies were not clear, was a a-synuclein transgene expressed alongside mito-dR in CHCHD2 mutant/deficiency background? A proper genotype labelling in the figure legends should be provided.
- 6) In Figure 4a, no. of total TH-positive neurons is straight forward, but what does no. of dR/TH positive neurons mean? Were dR positive neurons stained with anti-myc antibody and TH-positive neurons stained with anti-TH or were they GFP positive neurons if DdcGal4, GFP line was used? Genotypes should be provided here.
- 7) In Figure 4b, was the dopamine level measured from dissected brain extract or whole head extract? It is not clear from the Method section if the so called "brain extract" was from brain alone or whole head. This is important detail as it is well known that the eye pigment from the head could interfere with the dopamine peak in the HPLC analysis.
- 8) In Figure 4d, was RNAi UCP4A and mito-dR driven in CHCHD2^{-/-} background?
- 9) In Figure 4e, which motor behavior was described here, locomotion or flight? What is on the Y-axis? It does not tally with flying (Figure 4c) nor locomotion (Supplement Figure 4b). Also, is DdcGal4 a suitable driver for studying locomotion activity?

Point-by-point responses to reviewer comments

We greatly appreciate the feedback from the three reviewers on our manuscript (MS# COMMSBIO-19-0097A) titled “**Light-driven mitochondria relieve a *Drosophila* model of Parkinson’s disease**”. We found the comments to be highly constructive for improving our manuscript. We have performed additional experiments to address the questions and concerns of the reviewers and have included data showing that mito-dR elevates $\Delta\Psi_m$ in *dCHCHD2*^{-/-} flies using a mitochondrial potential indicator TMRM. Below are our point-by-point responses to the reviewers’ comments. The original comments by the reviewers are shown in blue. The literature cited here is listed in the **References for responses** section.

Reviewer #1 (Remarks to the Author):

Imai et al. have used an in vivo drosophila model to investigate the mitochondrial mechanism in a mitochondrial model of PD. Furthermore they use an optogenetic method (overexpression of a light driven proton pump) as a proof of principle to show that modulation of proton transfer/leak during respiration is 1) possible in living flies and 2) is able to alleviate certain PD phenotypes namely electron leak and ROS.

It is novel for several reasons: 1) The gene CHCHD2 is a relatively novel PD gene and little is known about the underlying biology 2) the work is performed in vivo 3) for the first time provide a mitochondrial mechanism that could be exploited as an entry point for therapy. Of course much more work is needed to understand how to translate this to the clinic.

The study is well designed. The strains of drosophila include WT, mutant and then with either functional DR or non-functional DR to prove that the effects are due to the light stimulation. It is not entirely clear whether these expression constructs are somatically expressed, organ specific or neuron specific. In the results some experiments appear to be in flies where the constructs are expressed in the whole body and then later in DA neurons. In the methods the drosophila genetics explains how they generated the flies. The manuscript would be improved by making clear which figures and sub-figures were using which flies.

For fly experiments, we employed a conventional *two-component* gene expression system (the so-called GAL4-UAS system) widely used for transgene expression in the whole body, specific tissues and specific cell populations (Brand and Perrimon, 1993). For the sake of clarity, we added a genotype list used here in the **Supplementary Information**.

Please give a more detailed description of how the calcium imaging was performed including an explanation of how the fly brains were dissected for the experiments.

We added a detailed description in the **Methods** section as follows (additional writing is highlighted in red):

Each fly head was held in a hole (1-mm in diameter) in a plastic plate (Fisherbrand, Cat. No. 12-547) with nail polish, and the mouth parts and the cuticles around the antennae were removed by tweezers. Ca²⁺ imaging of brain DA neurons expressing mito-GCaMP6 or GCaMP6f was carried out in Ca²⁺-free HL-3 using an Eclipse FNI microscope (Nikon) equipped with an electrical stimulation setup containing a SEN-3401 (Nihon koden) and SS-104J (Nihon koden). The lateral side of the antennal lobes was stimulated using a glass electrode (Warner Instruments, Cat. No. W3 64-0792) with a 100- μ m tip diameter made by an electrode puller (Sutter Instrument, P-97). GCaMP images were recorded for 1 min (1 frame/0.03 s), and 40 Hz electrical stimulation (5 V with 15-ms duration 10-ms intervals) for 1 s was performed in the antennal regions shown in Supplementary Fig. 3b 10 s after the recording. Each imaging was carried out within 10 min after dissection. Ca²⁺ imaging data were processed using NIS-Elements software (Nikon, Ver. AR-4.40.00), ImageJ-Fiji and Excel (Microsoft 2010).

To contribute to related research fields, we will publish the Ca²⁺ imaging method in Protocol Exchange or elsewhere.

It is not clear whether the statistical analysis refers to statistical n as sole number of flies and/or whether the experiments were repeated on different days or not. We are also concerned about the different numbers of flies per group (n=8-12). This could bias the statistics. Could the authors comment on their statistical approach?

We have clearly defined the meaning of 'n' in each figure legend. All data presented here were aggregates of different batches of flies, and experiments and sampling were repeated on different days. The behavioral analyses and the sampling for dopamine measurement were repeated in the same time window of different days using different batches of flies. For live-imaging analysis, we excluded samples that were not appropriately dissected, those from whom recordings were not obtained, and those that showed values over $2 \pm \text{SD}$, which made obtaining the sample sizes set by the experimental design difficult. We have now repeated the experiments to reach similar

sample sizes in Fig. 1d, e, f. Fig. 2b and d were re-examined because the previous Fig. 2b and d were measured by different experimenters, making comparisons between the values in Fig. 2b and d difficult. The addition of data did not affect the results except for those in Fig. 1e. Fig. 1e now suggests that mito-dR does not significantly stimulate ATP production in the cell bodies of dCHCHD2^{+/+} neurons effectively, which partly supports the observation that ATP production is not stimulated by mito-dR in the heads of dCHCHD2^{+/+} flies (Fig. 1d).

It is interesting to see that the CHCHD2 aged flies have significantly less dopamine but the light activation is not able to alleviate this effect significantly. Does this mean that the mitochondrial effect/damage has already been triggered and initiated a DA specific response? Could the authors comment/discuss why they think the DA phenotype is not altered by the light but the flight and TH neuron count effect is?

In insects, including *Drosophila*, dopamine is also synthesized in epidermal cells as well as dopaminergic neurons (Yamamoto and Seto, 2014). Dopamine produced by epidermal cells is oxidized into melanin and used for hardening of the cuticle (insect exoskeleton). As cuticle pigmentation and cuticle hardening are predominant in young flies just after eclosion, similarly, an age-dependent rapid reduction in dopamine in fly heads appears to be a physiological event, unlike in humans (Imai *et al.*, 2008). For dopamine measurement, we extracted dopamine from brain tissues, including epidermal cells and cuticles, due to the difficulties associated with their separation. Because dopamine data in 3-day-old flies would predominantly reflect epidermal dopamine production, we removed these data. Instead, we added data for dopamine measurement at 30 days old to Fig. 4b. An age-dependent decline in dopamine was noticeable regardless of genotype (please also refer to the previous data at 3 days old), and mito-dR did not overcome this dopamine reduction at 30 days old. However, brain dopamine contents may not accurately reflect the survival of dopaminergic neurons and their activity due to the contribution of epidermal cells to dopamine content.

It is also interesting that UCP4A seems to be involved in the protection mechanism. It would be very interesting to see whether UCP4A stimulation (or mild uncoupling) in combination with light stimulation could alleviate the DA deficit. Or at least the authors could allude to their hypothesis why mitochondrial targeted treatments may not rescue all DA specific phenotypes.

We expressed two different UCP4A transgenes (FlyORF and gift from Dr. T. Wang (Wu *et al.*, 2014)) along with mito-dR in a portion of dopaminergic (DA) neurons covered by the *Ddc-GAL4* driver under light irradiation conditions. Both UCP4A transgenes tended to rescue the loss of *dCHCHD2*^{-/-} DA neurons (PPM2 DA neuron number: 4.68 ± 0.20 in *dCHCHD2*^{-/-} DA, mito-dR NA with UCP4A [FlyORF] vs. 3.8 ± 0.41 in *dCHCHD2*^{-/-}, mito-dR NA without UCP4A; $p = 0.0558$ by two-tailed *t*-test), suggesting that mild uncoupling via UCP4A at least partly improved the survivability of *dCHCHD2*^{-/-} DA neurons. However, the coexpression of UCP4A with mito-dR WT reduced DA neuron survivability, considering that the normal number of PPM2 DA neurons covered by the *Ddc-GAL4* driver is approximately 6 (Please also refer to the upper graph in Fig. 4a). This result may suggest that increased uncoupling by UCPs or overexpression of a mitochondrial inner membrane protein such as UCP4A hampers mitochondrial function. We present the data with FlyORF as Fig. S4d because the results with the fly line from Dr. T. Wang was very similar to those obtained with FlyORF. The number of PPM2 DA-positive neurons in the fly line from Dr. T. Wang is shown below as a graph.

The numbers of dR-positive PPM2 cluster DA neurons in 30-day-old flies are graphed (mean \pm s.e.m.). $n = 22$ flies per genotype. N.S., not significant by Tukey-Kramer test.

Minor comments:

Concerning fly age. All flies were tested within a range 30 days or 14/15 days, which seems reasonable. In Figure 4B Imai et al investigate the effect of age at 3 and 15 days old. At least three time points to draw a linear relationship would greatly support the claim of age dependency. The authors refer to Figure 1F when discussing age dependency but here there is only one age group. It is confusing.

We added brain ATP data from 15-day-old flies to Fig. 1d (corresponding to the previous Fig. 1f). The brain ATP levels were similar among the four groups at 15 days old, suggesting that the brain ATP production in *dCHCHD2*-deficient flies is not hampered at this stage, and an age-dependent drop in ATP production was demonstrated.

As mentioned above, data of brain dopamine contents at 3 days of age were removed, and data of brain dopamine contents at 30 days were added in Fig. 4b.

Fig 1B: The pH scale bar is a bit confusing and an axis scale bar would be more helpful.

We changed the pH scale per the reviewer's suggestion.

Fig 1C: kDa missing for TIM23 and GAPDH Western blots. Fig 1D: kDa missing for CHCHD2 and actin.

We added the information on the molecular weight of these proteins in Fig. S1b and c (previous Fig. 1C and D).

For the box plots the lines defining the axes are missing. Formatting of the figures, especially the graphs could be improved.

We added the Y axis to the box plots and bar graphs. For better visibility, the X axis was omitted in some graphs, in which baselines are not critical. A similar formatting in box plots was applied in previous research papers (e.g., Guzman, Nature 2010).

Reviewer #2 (Remarks to the Author):

In this manuscript “ Light-driven mitochondria relieve a Drosophila model of Parkinson’s disease”, Imai et al., show that driving the activity of a mitochondria-targeted proton transporter (dR) using light, rescues some of the phenotypes associated with the loss of the PD-associated mitochondrial gene CHCHD2. Specifically, they show that light-induced activation of dR channel can, in some cases elevate ATP levels, as well as decrease mitochondrial peroxide production and restore Ca²⁺-buffering activity of mitochondria—all of which are defective in CHCHD2 null animals. In addition, they show that activation of mito-dR decreases a-synuclein load in DA neurons and restores DA production and motor activity in CHCHD2 flies.

The data are provocative and the manuscript is clearly written, although I had some issues with data presentation which I have outlined below. The authors convincingly show that activation of the dR channel in CHCHD2 null animals is phenotypically protective. However, data regarding the mechanism by which this protection is conferred are less compelling. It is difficult to understand which of the various changes that occur upon activation of dR are responsible for the protective effects, a problem the authors criticize in cultured cells (lines 66-67). In addition, there are conceptual issues as well as missing experimental controls that are needed to convincingly make their point regarding the mechanism of dR activity-dependent rescue in DA neurons.

Major comments:

1) A major question throughout, is why dR expression has different effects in the CHCHD2^{+/+} flies versus the CHCHD2^{-/-} flies. If dR activation does indeed act through enhancing Δp in vivo (which is not shown), why do we not see any effects of this—either effects from the increased Δp , or effects due to adaptation to prolonged

increases in Δp --in the control animals? This question is evident in multiple cases such as Figure 1f, where activation of dR in *dCHCHD2* ^{+/+} animals has no effect on net ATP levels, Figure 3 where dR activation has no effect on Ca^{2+} in *dCHCHD2* ^{+/+} animals. This needs to be addressed experimentally, or at the very least, discussed extensively.

Because direct measurements of Δp in flies are difficult to obtain, we estimated the mitochondrial membrane potential ($\Delta\Psi_m$) in motor neuron terminals using the $\Delta\Psi_m$ indicator TMRM. The results indicated that mito-dR did not increase the $\Delta\Psi_m$ further in *dCHCHD2* ^{+/+} flies (new Fig. 1c). We also repeated ATP measurements in fly heads as in Fig. 1d (corresponding to the previous Fig. 1f) by different experimenters and confirmed the reproducibility. Thus, we believe that an overly polarized state does not physiologically occur in mitochondria. Supporting this idea, inhibition of F_0F_1 ATPase by oligomycin did not prominently increase $\Delta\Psi_m$ in human cultured cells (Zulian *et al.*, 2014). The phenomenon could be explained by uncoupling or proton leak mediated by uncoupling proteins, the adenine nucleotide translocase, the glutamate carriers and the permeability transition pore complex (Amoedo *et al.*, 2016; Azzu and Brand, 2010). In Fig. 3a, *dCHCHD2* ^{+/+} flies expressing mito-dR WT showed a tendency toward a rapid response in terms of $[Ca^{2+}]_m$ efflux compared to *dCHCHD2* ^{-/-} flies expressing mito-dR WT or *dCHCHD2* ^{+/+} flies expressing mito-dR NA (Fig. S3). This difference might be derived from the effects of prolonged uncoupling by the Na^+/H^+ exchanger (which could potentially activate Na^+/Ca^{2+} exchanger) or the activation of a putative Ca^{2+}/H^+ exchanger, such as *Letm1*, in the energized state.

We discussed this issue, including the effect on the Ca^{2+} dynamics in the **Discussion** section, as follows:

*In contrast, mito-dR WT did not increase $\Delta\Psi_m$ or ATP production in *dCHCHD2* ^{+/+} flies, suggesting that an overly polarized state does not physiologically occur in mitochondria (Fig. 1c-e). Supporting this idea, the inhibition of F-type ATPase complex V by oligomycin did not prominently increase $\Delta\Psi_m$ in human cultured cells (Zulian *et al.*, 2014). This observation could be explained by uncoupling or proton leak mediated by*

UCPs, the adenine nucleotide translocase, the glutamate carriers and the permeability transition pore complex (Amoedo et al., 2016; Azzu and Brand, 2010). dCHCHD2^{+/+} flies expressing mito-dR WT exhibited a tendency toward a rapid [Ca²⁺]_m efflux compared to dCHCHD2^{-/-} flies. This effect might be due to prolonged uncoupling by the Na⁺/H⁺ exchanger (which could potentially activate the Na⁺/Ca²⁺ exchanger) or the activation of a putative Ca²⁺/H⁺ exchanger such as Letm1 in the energized state (Haumann et al., 2018; Jiang et al., 2009). These altered Ca²⁺ dynamics in DA neurons might reflect the behavioral changes observed in dCHCHD2^{+/+} flies.

2) Figure 1f: There is no significant change in ATP production in dCHCHD2 ^{+/+} whole flies as measured by ATP/protein. However, when expressed under a DA driver (Fig 1g), there is an increase in FRET signal in Ateam. These two results are inconsistent and should be reconciled experimentally.

Our new data on the terminals of abdominal motor neurons (Fig. 1b) indicated that mito-dR does not increase $\Delta\Psi_m$ in *dCHCHD2^{+/+}* flies. In addition, we repeatedly measured ATP production in the cell bodies and terminals of dopaminergic (DA) neurons (Fig. 1e and f) in response to a comment from Reviewer 1. The data indicated that ATP production was increased in neuronal terminals but not in cell bodies, although there was still a tendency toward increased ATP production in the cell bodies of *dCHCHD2^{+/+}* DA neurons. Thus, mito-dR appears to be functional in specific structural components (such as neuronal terminals) but not in the entire brain, potentially due to the availability of light and differing cell types.

3) Figure 2e: Why is ROS not decreased in wt (*dCHCD2^{+/+}*) animals?

Little is known of the mechanism regulating redox changes in cells or tissues of living animals. For instance, age-independent heterogeneity in mitochondrial H₂O₂ levels has been reported in fat bodies as an enigmatic phenomenon (Albrecht *et al.*, 2011). While H₂O₂ from mitochondria is thought to be the major oxidant species in the brain (Stepanova *et al.*, 2019), auto-oxidation from environmental conditions such as oxygen

in the air and light and age-dependent impairment in redox regulation and inflammation might cause oxidation of brain lipids and the roGFP2 biosensor as basal oxidant factors. Given that the oxidative status of roGFP2 in *dCHCHD2*^{+/+} flies is basal from nonmitochondrial factors, the results are reasonable.

We discussed this issue in the **Discussion** section as follows (additional writing is highlighted in red):

*In addition, mito-dR suppressed oxidative stress derived from mitochondria, which is likely due to uncoupling mediated by proton leakage where UCP4A is involved (Fig. 4f) (Guzman et al., 2010; Jastroch et al., 2010; Wu et al., 2014). Consistent with this observation, the elevation in brain lipid oxidation by CHCHD2 loss was successfully suppressed by mito-dR (Fig. 2a-d). However, brain lipid oxidation in normal flies (*dCHCHD2*^{+/+}) was not further reduced. A similar situation was also observed in the cell bodies of DA neurons (Fig. 2e). While H₂O₂ from mitochondria is thought to be the major oxidant species in the brain (Stepanova et al., 2019), auto-oxidation from environmental conditions such as oxygen in the air, short wavelength light, age-dependent impairment in redox regulation and inflammation might cause oxidation of brain lipids and the roGFP2 biosensor as basal oxidant factors. In addition, we observed a subset of glial cells with mildly elevated mitochondrial H₂O₂ production when mito-dR was activated in these cells (Supplementary Fig. 1e). These factors may contribute to a null effect of mito-dR in *dCHCHD2*^{+/+} flies when mito-dR is ubiquitously expressed.*

4) Figure 3a: Changes in [Ca²⁺]_c and [Ca²⁺]_m in *dCHCHD2*^{+/+} expressing mito-dR needs to be shown. Is the rescue back to wild-type levels?

We added data on *dCHCHD2*^{+/+} flies expressing mito-dR WT in Fig. 3a. The pattern of Ca²⁺ dynamics in *dCHCHD2*^{-/-} flies with mito-dR WT was more similar to that in *dCHCHD2*^{+/+} flies with mito-dR WT than that in *dCHCHD2*^{-/-} flies with mito-dR NA.

5) Line 177: α -synuclein aggregation in CHCHD2 $-/-$ animals. Data should be shown.

A manuscript showing prominent α -synuclein accumulation in a PD patient with a CHCHD2 mutation and fly model has been submitted elsewhere (under revision). We attached the manuscript as supporting data for the review.

6) Figure 3b: Are total α -synuclein levels in CHCHD2 $-/-$ and CHCHD2 $+/+$ animals expressing mito-dR, equivalent?

We used the same α -synuclein transgenic and GAL4 driver lines and adjusted the copy number of UAS transgenes between the two genotypes so as not to affect GAL4 titration (Please also refer to the genotype list in **Supplementary Information**). Under this condition, total α -synuclein was also increased in *dCHCHD2*^{-/-} flies, probably due to defects in autophagy, as shown in the figure below (Supporting data are also included in the above manuscript).

Figure legend:

SDS sample buffer-soluble lysate prepared from 6 30-day-old male fly heads was western blotted with the indicated antibodies. Two independent results are shown. The genotypes used here were as follows:

+/Y; UAS-LacZ/+; Ddc-Gal4/+ (dCHCHD2 +; LacZ +)

CG5010^{null}/Y; UAS-LacZ/+; Ddc-Gal4/+ (dCHCHD2 -; LacZ +)

+/Y; UAS- α -Synuclein LP2/+; Ddc-Gal4/+ (dCHCHD2 +; α -Synuclein +)

CG5010^{null}/Y; UAS- α -Synuclein LP2/+; Ddc-Gal4/+ (dCHCHD2 -; α -Synuclein +)

CG5010 indicates dCHCHD2.

7) Are changes in ATP production, lipid peroxidation and axonal ROS all occurring in the same neuronal populations in dCHCD2^{-/-} animals? If so, this needs to be better indicated?

We monitored H₂O₂ production (Fig. 2e, f) and performed Ca²⁺ imaging (Fig. 3a) in the

same DA neuron population (PAM DA neurons) used for the assessment of ATP production (Fig. 1e, f). We describe the use of PAM DA neurons in the corresponding legends. For lipid peroxidation (Fig. 2a-d), estimating the degree of lipid peroxidation in specific neuronal populations is technically challenging; therefore, we evaluated it in the whole brain. Details are described in the legends.

8) Figures are difficult to read and sloppily put together. For instance, in Figure 3b it is difficult to see what the bar and whisker plots correspond to.

We modified the figures for better presentation.

In **Statistical analysis** section of the **Methods**, we inserted a sentence to explain the whisker plots.

Boxes of whisker plots indicate the 25th to 75th percentiles, horizontal lines in the boxes indicate the 50th percentile, and whiskers represent the maximum and minimum values.

Minor comments:

1) Figure 2d: Does expression of the channel affect 4-HNE levels? 4-HNE levels look lower in dCHCD2 ^{-/-} expressing NA vs those expressing mito-dR.

We performed the 4-HNE experiment again because the previous 4-HNE data for mito-dR WT and NA were measured by different experimenters due to handover. New data in Fig. 2d suggest that mito-dR NA does not lower 4-HNE levels.

Reviewer #3 (Remarks to the Author):

This manuscript is original because it characterizes a novel reagent in which a light-dependent proton transporter was targeted specifically at the mitochondria to enhance its function. In addition, this manuscript also demonstrated the successful use of several optogenetic reagents such as ATP (Ateam), ROS (roGFP2-Orp1) and Ca²⁺ (mito-GCaMP6).

This is an interesting study, with some significant observations and conclusions. However, it is poorly written and overly complicated, paradoxically, even though both the text and figures are complicated, the work is not described in sufficient detail.

As mentioned above the manuscript is poorly written. Specifically, it lacks attention to details, hence making reading difficult. Abbreviations were introduced abruptly in the Introduction, for example: $\Delta\Psi_m$, Δp and UCPs.

We spelled out $\Delta\Psi_m$, Δp and UCPs in the first appearance of the **Introduction**. We carefully modified the manuscript to concisely explain the significance of our findings and added the details in the **Methods** and **Supplementary Information** so as not to disrupt the overall flow of the manuscript.

The lack of attention to detail is most obvious in describing experimental designs and data presentation. Specifically, genotypes were not presented in the Figures, Figure Legends nor explained in the text. For example: detection of a-synuclein accumulation in mito-dR expression in dCHCHD2 mutant background (Figure 3b, c), in nowhere was UAS a-synuclein expression described.

We have added the genotype list to the **Supplementary Information** and additional descriptions in the legends to show the experimental designs clearly.

Overall, the characterisation of mito-dR transgenic lines is insufficient. It also lacks experimental detail on how mito-dR WT and NA transgenic lines were generated. The reader could only assume they were UAS lines.

The myc-mito-dR gene has been reported in a previous study that is cited in this manuscript (Hara *et al.*, 2013). The construction of plasmids for transgenes and the generation of transgenic flies are described in the **Methods** section as follows:

Antibodies and plasmids

.....A complementary DNA fragment of myc-mito-dR in the pCMV (Hara *et al.*, 2013) vector was subcloned into the EcoRI and XbaI sites of the pUAST vector to generate mito-dR WT. The mito-dR NA mutant was generated using QuikChange site-directed mutagenesis kit (Agilent Technologies).

Drosophila genetics

.....Transgenic lines carrying mito-dR or mito-GCaMP6 were generated on the w^{1118} background (BestGene).

For further characterization of mito-dR, we also examined the effects of mito-dR in glial cells and found that mito-dR was functional in the cortex and subperineurial glia covered by *NRV2-GAL4* in terms of ATP production. Surprisingly, the expression of mito-dR WT but not NA by the pan glia driver *repo-GAL4* caused a shorter lifespan (all flies died by 20 days old). Thus, the response to the enhancement of Δp in mitochondria and the sensitivity to mito-dR expression appear to be different among cell types. Interestingly, mito-dR stimulated H₂O₂ production in the cortex and subperineurial glia, which suggests that glial mitochondria differ from those of dopaminergic or glutamatergic motor neurons and/or that heterogeneity in mitochondrial H₂O₂ levels in tissues exists as reported (Albrecht *et al.*, 2011).

The author should demonstrate if light-activated mito-dR has an effect on $\Delta\Psi_m$, with a mitochondrial potential indicator, Tetramethylrhodamine, methyl ester (TMRM).

Adult abdominal motor neuron terminals were well irradiated by light. We estimated the effects of mito-dR on the mitochondrial membrane potential ($\Delta\Psi_m$) using TMRM and confirmed that $\Delta\Psi_m$ was elevated in *dCHCHD2*^{-/-} motor neuron terminals (new Fig. 1c).

Specific comments:

1) In Figure 1d, the author should specify which tissue or if the whole flies were used in the Western blot.

We have added the following description to the legend (The former Fig. 1d was moved to Fig. S1c):

The expression of mito-dR on wild-type (*w*¹¹¹⁸) and *dCHCHD2*^{-/-} backgrounds was confirmed using brain tissues.

2) In Figure 1f, were the brain ATP contents extracted from whole heads or brain only?

We generally used whole heads because brain dissection causes variability in the results. We changed the description of ‘brain ATP contents’ to ‘whole-head ATP contents.’ The former Fig. 1f was moved to Fig. 1d.

3) Figure 1g is poorly described. Was YFP/CFP ratio measured in cell bodies of PAM cluster of DA neurons?

We corrected the legends as follows (The former Fig. 1g was moved to Fig. 1e):

Measurement of relative ATP levels (the ratio of 510-580 nm/440-510 nm emission fluorescence intensity) in the cell bodies (g) and nerve terminals (h) of PAM cluster DA neurons in 30-day-old males expressing mito-dR WT or NA using the fluorescence ATP biosensor ATeam.

4) In Figure 2e, f, and Supplement Figure 3, was it a typo, PAM neurons instead of DAM?

We corrected the typos.

5) In Figure 3, genotypes of flies were not clear, was a a-synuclein transgene expressed alongside mito-dR in CHCHD2 mutant/deficiency background? A proper genotype labelling in the figure legends should be provided.

We added an explanation of the experimental design in the Fig. 3b legend and a genotype list in the **Supplementary Information**.

6) In Figure 4a, no. of total TH-positive neurons is straight forward, but what does no. of dR/TH positive neurons mean? Were dR positive neurons stained with anti-myc antibody and TH-positive neurons stained with anti-TH or were they GFP positive neurons if DdcGal4, GFP line was used? Genotypes should be provided here.

As shown in Fig. S4a, the *Ddc-GAL4* driver covers most PPM2 neurons but not all TH-positive neurons. A similar situation was also observed with the *TH-GAL4* driver (Whitworth *et al.*, 2005). Thus, we counted the anti-Myc (dR)/anti-TH-positive (upper graph) and anti-TH-positive (lower graph) neurons to estimate the effects of dR expression on DA neuron survival.

7) In Figure 4b, was the dopamine level measured from dissected brain extract or whole

head extract? It is not clear from the Method section if the so called “brain extract” was from brain alone or whole head. This is important detail as it is well known that the eye pigment from the head could interfere with the dopamine peak in the HPLC analysis.

We used head extracts in the experiments and corrected the description in the **Methods** section. Although we also recognize the issue of eye pigment, optimization of the mobile phase resolved the problem.

8) In Figure 4d, was RNAi UCP4A and mito-dR driven in CHCHD2^{-/-} background?

We added the appropriate labels in Fig. 4d and e. A genotype list was also added to the **Supplementary Information**.

9) In Figure 4e, which motor behavior was described here, locomotion or flight? What is on the Y-axis? It does not tally with flying (Figure 4c) nor locomotion (Supplement Figure 4b). Also, is DdcGal4 a suitable driver for studying locomotion activity?

Fig. 4e shows the data from the startle-induced negative geotaxis assay (climbing assay) to estimate motor functions. We added a description of the Y-axis in the graph. Locomotion in Supplementary Figure 4b was assessed to measure spontaneous movements, which were automatically monitored by the DAM system. Startle-induced locomotion is regulated by PAM DA neurons, which are covered by the *Ddc-GAL4* driver (Riemensperger *et al.*, 2013).

References for responses

Albrecht, S.C., Barata, A.G., Grosshans, J., Teleman, A.A., and Dick, T.P. (2011). In vivo mapping of hydrogen peroxide and oxidized glutathione reveals chemical and regional specificity of redox homeostasis. *Cell Metab* 14, 819-829.

Amoedo, N.D., Punzi, G., Obre, E., Lacombe, D., De Grassi, A., Pierri, C.L., and Rossignol, R. (2016). AGC1/2, the mitochondrial aspartate-glutamate carriers. *Biochimica et biophysica acta* 1863, 2394-2412.

Azzu, V., and Brand, M.D. (2010). The on-off switches of the mitochondrial uncoupling proteins. *Trends Biochem Sci* 35, 298-307.

Brand, A.H., and Perrimon, N. (1993). Targeted Gene-Expression as a Means of Altering Cell Fates and Generating Dominant Phenotypes. *Development* 118, 401-415.

Guzman, J.N., Sanchez-Padilla, J., Wokosin, D., Kondapalli, J., Ilijic, E., Schumacker, P.T., and Surmeier, D.J. (2010). Oxidant stress evoked by pacemaking in dopaminergic neurons is attenuated by DJ-1. *Nature* 468, 696-700.

Hara, K.Y., Wada, T., Kino, K., Asahi, T., and Sawamura, N. (2013). Construction of photoenergetic mitochondria in cultured mammalian cells. *Scientific reports* 3, 1635.

Haumann, J., Camara, A.K.S., Gadicherla, A.K., Navarro, C.D., Boelens, A.D., Blomeyer, C.A., Dash, R.K., Boswell, M.R., Kwok, W.M., and Stowe, D.F. (2018). Slow Ca(2+) Efflux by Ca(2+)/H(+) Exchange in Cardiac Mitochondria Is Modulated by Ca(2+) Re-uptake via MCU, Extra-Mitochondrial pH, and H(+) Pumping by FOF1-ATPase. *Front Physiol* 9, 1914.

Imai, Y., Gehrke, S., Wang, H.Q., Takahashi, R., Hasegawa, K., Oota, E., and Lu, B. (2008). Phosphorylation of 4E-BP by LRRK2 affects the maintenance of dopaminergic neurons in *Drosophila*. *The EMBO journal* 27, 2432-2443.

Jastroch, M., Divakaruni, A.S., Mookerjee, S., Treberg, J.R., and Brand, M.D. (2010). Mitochondrial proton and electron leaks. *Essays Biochem* 47, 53-67.

Jiang, D., Zhao, L., and Clapham, D.E. (2009). Genome-wide RNAi screen identifies Letm1 as a mitochondrial Ca²⁺/H⁺ antiporter. *Science* 326, 144-147.

Riemensperger, T., Issa, A.R., Pech, U., Coulom, H., Nguyen, M.V., Cassar, M., Jacquet, M., Fiala, A., and Birman, S. (2013). A single dopamine pathway underlies progressive locomotor deficits in a *Drosophila* model of Parkinson disease. *Cell Rep* 5, 952-960.

Stepanova, A., Konrad, C., Manfredi, G., Springett, R., Ten, V., and Galkin, A. (2019). The dependence of brain mitochondria reactive oxygen species production on oxygen level is linear, except when inhibited by antimycin A. *Journal of neurochemistry* 148, 731-745.

Whitworth, A.J., Theodore, D.A., Greene, J.C., Benes, H., Wes, P.D., and Pallanck, L.J. (2005). Increased glutathione S-transferase activity rescues dopaminergic neuron loss in a *Drosophila* model of Parkinson's disease. *Proc Natl Acad Sci U S A* 102, 8024-8029.

Wu, K., Liu, J., Zhuang, N., and Wang, T. (2014). UCP4A protects against mitochondrial dysfunction and degeneration in pink1/parkin models of Parkinson's disease. *FASEB journal* :

official publication of the Federation of American Societies for Experimental Biology 28, 5111-5121.

Yamamoto, S., and Seto, E.S. (2014). Dopamine dynamics and signaling in *Drosophila*: an overview of genes, drugs and behavioral paradigms. *Exp Anim* 63, 107-119.

Zulian, A., Tagliavini, F., Rizzo, E., Pellegrini, C., Sardone, F., Zini, N., Maraldi, N.M., Santi, S., Faldini, C., Merlini, L., *et al.* (2014). Melanocytes from Patients Affected by Ullrich Congenital Muscular Dystrophy and Bethlem Myopathy have Dysfunctional Mitochondria That Can be Rescued with Cyclophilin Inhibitors. *Front Aging Neurosci* 6, 324.

REVIEWERS' COMMENTS:

Reviewer #1 (Remarks to the Author):

Imai et al. have responded to the comments, made substantial changes to the manuscript including those to the text and figures and Imai et al. also provided more experimental data. Most of my concerns were due to the presentation of the data in the figures and remaining questions regarding statistics. I am now satisfied with the revised manuscript and think this interesting work is now presented well.

My broader questions were answered and the method section was improved. The new discussion section was also improved.

The paper shows evidence that improving mitochondrial function in PD could be useful for therapeutics and provides a new method for mitochondrial research.

Reviewer #3 (Remarks to the Author):

The revised manuscript entitled "Light-driven mitochondria relieve a Drosophila model of Parkinson's disease" by Imai et al., has improved from the sloppiness of the original version. It is also much easier to read and follow.

The authors have reacted positively to the constructive comments from reviewers. The authors have taken the criticism of "the lack of attention to detail in describing experimental designs and data presentation" seriously, and has presented the manuscript in a more thorough manner. For example, the provision of a list of all genotypes helps readers to understand the genetic background of the animal used in the experiments.

In addition to the more well written manuscript, the authors have also conducted new experiments and hence not only explaining doubt casted by reviewers but also enhance the validity of the results obtained and conclusion drawn. For example: the authors addressed concern of reviewer 2 and 3 regarding effect of enhancing Δp on CHCHD2 wild type or mutant flies by estimating the mitochondrial membrane potential ($\Delta\Psi_m$) in motor neuron terminals using the $\Delta\Psi_m$ indicator TMRM. This new data provided explanation for the comments by both reviewers.

Overall, this manuscript is very much improved and I would recommend it for publication in Communications Biology